# From Welfare to Utility: Generalized Objectives in Budget-Feasible Procurement

Alon Eden [1]   Kira Goldner [2]   Eldar Kerner [1]   Thodoris Tsilivis [2]

## Abstract

We study mechanism design for the budget-feasible procurement problem, a natural problem that arises when a buyer wants to procure goods or services from multiple strategic sellers who each have a cost to provide that service, the buyer has a value for each service procured, but is constrained by a budget. In contrast to prior work, which has focused on buyer value maximization for this problem, we solve for optimal and approximately-optimal mechanisms for the objectives of buyer utility (value of procured services minus payments), welfare (value minus production costs), and generalizations of the two. For welfare, we design a simple mechanism that obtains a constant-factor approximation for the prior-free (worst-case) setting. As prior-free mechanisms fail to provide any guarantee for utility, even for a single seller, we consider Bayesian settings, where the buyer has distributional knowledge over sellers' costs. We first provide a utility-optimal mechanism that satisfies the buyer's budget constraint *in expectation*, then we show how to modify the mechanism to satisfy the budget constraint *ex-post*, for every realization of seller costs, while still obtaining near-optimal utility guarantees. Finally, we generalize our mechanisms to other objectives.

## 1. Introduction

Consider a company that wishes to procure compute from some of $n$ possible sellers. The buyer has some value $v_i$ for each compute contract that they procure depending on the seller $i$. However, each seller also has their own private cost $c_i$ to supply compute—e.g., due to electricity, hardware usage, and system maintenance. Finally, the buyer has a budget $B$ that they have set aside for spending on compute, and the sum of payments that they make to the sellers cannot exceed this budget.

This is the budget-feasible procurement problem, relevant in any scenario where a buyer seeks to procure goods or services from self-interested sellers and is constrained by a budget. Such a scenario arises in many settings relevant in today's modern computing ecosystem:

- **Online crowdsourcing task:** a requester in Mechanical Turk seeks to procure taskers subject to some budget (Anari et al., 2014; Balkanski & Hartline, 2016; Singla & Krause, 2013; Singer & Mittal, 2013).

- **API access to language models:** a user wants to access specialized language models through an Application Programming Interface (API) to perform a task, where every access to the API is monetized by the company training the language model (Bhawalkar et al., 2025a).

- **Distributed/Federated learning:** a model trainer wants to purchase models trained locally by users on their private data in order to train a global model (Zhang et al., 2021).

This problem was first introduced by Singer (2010) and has since been extensively studied in algorithmic mechanism design, receiving particular attention from researchers at large technology platforms (Deng et al., 2025; Anari et al., 2018; Bei et al., 2012; Bhawalkar et al., 2025a; Chen et al., 2011). However, the vast majority of work focuses only on the objective of *value maximization*. The justification is that the buyer has set aside the budget to procure this service, and so there is no harm in using the full budget, only in maximizing the buyer's value given what is spent.

This paper questions that assumption by studying other well-motivated objectives, such as welfare and utility maximization. In many settings, unused budget could be reused elsewhere, making *buyer utility*—buyer value minus payments—a more appropriate objective. For instance, when the value of procured services represents the additional revenue the business can obtain from procuring these services, then the buyer's utility is the actual profit of the buyer, subject to their liquidity. This assumption is prevalent in the standard forward auction setting (1 seller, $n$ buyers), where it is assumed that a budgeted buyer wants to maximize their utility

---

[1]Hebrew University [2]Boston University. Correspondence to: Thodoris Tsilivis <tsilivis@bu.edu>.

*Proceedings of the 43rd International Conference on Machine Learning*, Seoul, South Korea. PMLR 306, 2026.

*subject to budget constraints* (Borgs et al., 2005; Dobzinski et al., 2008; Bhattacharya et al., 2010). This requires more careful consideration to trade off the value that the buyer receives with the money spent procuring the services. A second natural objective is *social welfare*, defined as the buyer's value minus the sellers' costs of the allocation. This objective is particularly relevant when the buyer's value benefits society (e.g., economically) and the sellers' costs capture societal harms, such as the environmental costs of compute. Moreover, while the procurer individually cares about their utility, they may instead choose to maximize the system's welfare for the longevity of the system, to ensure that sellers stay in the system and they can continue to earn utility from them.

In addition, we look at generalizations of these objectives. We consider a generalization of social welfare where seller costs are weighted by a parameter $\alpha$ which serves as a knob for social cost concerns: setting $\alpha = 0$ represents pure value maximization, as in prior works, while $\alpha \in (0, 1)$ represents an intermediate regime where value is prioritized but some weighted cost consideration is given, and $\alpha > 1$ represents a cost-aware regime, where societal harms are given extra consideration. Additionally, we consider a generalization of the utility objective where each agent's value is weighted by $a_i \geq 0$ and their payment by $b_i \geq 0$. We can retrieve the pure value maximization objective by setting $a_i = 1$ and $b_i = 0$, while increasing $b_i$ captures regimes with greater concern for the implications of payments.

Due to their great relevance, both the utility (Cary et al., 2008) and welfare (Deng et al., 2025) objectives have been studied in the procurement auction setting already, only without budget constraints. The goal of this paper is to solve the budget-feasible procurement problem for welfare, utility and a variety of other well-motivated objectives.

**Our Results.** We study the budget-feasible procurement problem for welfare, utility, and generalizations thereof. First, in Section 3, we investigate the prior-free setting of the original (Singer, 2010), that is, we maximize our objective over worst-case inputs of seller costs. We provide a simple mechanism that obtains a $\frac{1}{2+\sqrt{2}}$-approximation to the optimal social welfare. We show that this extends for various objectives such as maximizing value minus $\alpha$ times production costs for $\alpha \geq 0$, as well as some concave function of such objectives, representing decreasing marginal gains of accumulated welfare. Then, we turn to the larger technical challenge of utility maximization; however, no guarantee for utility is possible in the prior-free setting (Proposition 6).

Thus to maximize utility, in Section 4, we investigate the Bayesian setting, where seller costs are drawn from a known prior distribution. In Section 4.1, we solve the optimization

problem optimally when the buyer's budget constraint is soft, and need only be met in expectation over the realization of seller costs. In Section 4.2, we investigate the more challenging problem when the buyer's budget constraint must be satisfied for every realization of seller costs. We use our solution from Section 4.1 to construct our main result: a simple posted pricing mechanism that guarantees a $\frac{1}{5}$-approximation. Finally, in Section 4.3, we illustrate how to expand the above to handle more difficult prior distributions via "ironing," or forcing monotonicity, and extend this framework to more general objectives that are convex combinations of value and payment.

*Technical Contribution.* Our prior-free analysis for welfare parallels that of the approach introduced in (Papadimitriou & Singer, 2010; Chen et al., 2011) for value maximization when the buyer's valuation is additive. A key idea in their mechanisms is to construct a high-value set of sellers $W$ using conditions on the value-per-cost ratio of the agents, which ensures budget-feasibility. Our main challenge is in showing that, even though the condition of adding sellers to $W$ is based on their value, which can be arbitrarily larger than their associated welfare, the set $W$ has strong welfare guarantees as well.

Our Bayesian analysis for utility parallels that for value of Balkanski & Hartline (2016) in many places. However, their ex-post solution requires a "$k$-large market" assumption, i.e., it only considers when all sellers have small enough ex-ante posted prices that they consume at most $1/k$-th of the budget. Instead, we produce a general constant-factor approximation by using their $k$-large market approach when the smaller sellers produce a good fraction of the optimal utility, and contributing a different mechanism when the majority of utility comes from the large sellers. Note, however, that in using their $k$-large market analysis (Lemma 2), we discover a minor hole in their analysis. Our solution patches their hole, and also fixes a slight error with budget-feasibility. (See details in Remark 1.)

The most related literature is mentioned above and throughout our analysis where there are similarities, but we include an expansive review of related work in Appendix A.

## 2. Model and Preliminaries

In our model, we have a single buyer with expenditure (or budget) of $B \in \mathbb{R}_{\geq 0}$ and a set of $n$ sellers (henceforth, the agents) the buyer wants to purchase services from the sellers, subject to the buyer's budget constraints. Each seller can produce a (single) service at cost $c_i \in \mathbb{R}_{\geq 0}$, privately known to the seller, and the buyer values each service at a known $v_i \in \mathbb{R}_{\geq 0}$. The buyer has an additive value over services, that is, when procuring a set $S \subseteq [n]$ of services, they obtain value of $\sum_{i \in S} v_i$. The buyer wants to devise DSIC-

IR mechanisms to procure services from agents, in order to maximize various objectives subject to their expenditure constraints, as described below.

For a vector $\mathbf{t} = (t_1, \ldots, t_n)$, we denote by $\mathbf{t}_{-i}$ the vector when excluding the $i$th coordinate, and by $(\mathbf{t}_{-i}, \hat{t}_i)$ the vector that inserts $\hat{t}_i$ at the $i$th coordinate. A mechanism receives the *reported* production costs of sellers $\mathbf{c} = (c_1, \ldots, c_n)$, and returns a (possibly fractional) allocation vector $\mathbf{x} : \mathbb{R}_{\geq 0}^n \to [0, 1]^n$ and a payment vector $\mathbf{p} : \mathbb{R}_{\geq 0}^n \to \mathbb{R}_{\geq 0}^n$, $x_i(\mathbf{c})$ and $p_i(\mathbf{c})$ return $i$th coordinate of $\mathbf{x}(\mathbf{c})$ and $\mathbf{p}(\mathbf{c})$, respectively. The utility of a seller with cost $\hat{c}_i$ for reports $\mathbf{c}$ in mechanism $(\mathbf{x}, \mathbf{p})$ is

$$u_i(\hat{c}_i, \mathbf{c}) = p_i(\mathbf{c}) - \hat{c}_i \cdot x_i(\mathbf{c}).$$

We next define our incentive guarantees.

**Definition 1** (DSIC-IR). Consider a mechanism $\mathcal{M} = (\mathbf{x}, \mathbf{p})$.

- $\mathcal{M}$ is Individually Rational (IR) if for every $i$ and $\mathbf{c} = (c_1, \ldots, c_n)$, $u_i(c_i, \mathbf{c}) \geq 0$, that is, the utility of a seller from reporting their true cost is non-negative.

- $\mathcal{M}$ is Dominant-Strategy Incentive Compatible (DSIC) if for every $i$, $\mathbf{c} = (c_1, \ldots, c_n)$, and $\hat{c}_i$, $u_i(c_i, \mathbf{c}) \geq u_i(c_i, (\mathbf{c}_{-i}, \hat{c}_i))$, that is the agent cannot obtain a higher utility from reporting a different cost than their true cost.

A mechanism that is both DSIC and IR is said to be DSIC-IR.

In order to devise DSIC-IR mechanisms we use Myerson's canonical characterization of such mechanisms, adapted to our setting. An allocation function $\mathbf{x}$ is called *implementable* if there exist payments $\mathbf{p}$ such that $(\mathbf{x}, \mathbf{p})$ is DSIC-IR.

**Lemma 1** (Myerson's Lemma (Myerson, 1981)). *An allocation function $\mathbf{x}$ is implementable if and only if $x_i(\cdot, \mathbf{c}_{-i})$ is monotonically non-increasing for every $\mathbf{c}_{-i}$, and payments are given by the following formula:*

$$p_i(\mathbf{c}) = c_i \cdot x_i(\mathbf{c}) + \int_{c_i}^{\infty} x_i(z, \mathbf{c}_{-i}) \, dz. \tag{1}$$

Consider the special case of deterministic monotone allocation functions (that is $x_i(\mathbf{c}) \in \{0, 1\}$). In this case, it is easy to show that the payments are simply threshold payments, and the following holds.

**Observation 1.** *Consider a deterministic and monotone $\mathbf{x}$, if for some $\mathbf{c} = (c_1, \ldots, c_n)$, $x_i(\mathbf{c}) = 0$, then when using Myersonian payments as defined in Eq. (1), for every $\hat{c}_i$, $p_i(\hat{c}_i, \mathbf{c}_{-i}) \leq c_i$.*

In the Bayesian setting, in order to use virtual cost formulation of the objective, we consider the appropriate definition of regularity for procurement auctions.

**Definition 2** (Regularity). Let $F_i$ be the cumulative distribution function of agent $i$'s cost with density $f_i$. The distribution $F_i$ is said to be regular if the associated virtual cost function $\phi_i(c_i)$ is monotonically non-decreasing in $c_i$:

$$\phi_i(c_i) = c_i + \frac{F_i(c_i)}{f_i(c_i)}. \tag{2}$$

**Budget feasibility.** The buyer aims to procure items from a set of agents subject to a budget constraint $B$. Let $\mathbf{c} = (c_1, \ldots, c_n)$ denote as a vector of the reported costs, and let $p_i(\mathbf{c})$ denote the payment made to agent $i$. There are two main definitions, Ex-Post (Hard) and Ex-Ante (Soft), distinguishing by the strictness of the constraints.

**Definition 3** (Ex-Post Budget Feasibility). A mechanism $\mathcal{M}$ is Ex-Post Budget Feasible (or satisfies a Hard Budget Constraint) if, for every possible reported costs $\mathbf{c}$, the total payments never exceed the budget $B$: $\sum_{i=1}^{n} p_i(\mathbf{c}) \leq B$.

**Definition 4** (Ex-Ante Budget Feasibility). A mechanism $\mathcal{M}$ is Ex-Ante Budget Feasible (or satisfies a Soft Budget Constraint) if the expected total payment, taken over the joint probability distribution of the agents' costs $D$, does not exceed the budget $B$: $\mathbb{E}_{\mathbf{c} \sim D} \left[ \sum_{i=1}^{n} p_i(\mathbf{c}) \right] \leq B$.

**Objective Functions.** Consider an instance with sellers' costs $\mathbf{c}$ and values $\mathbf{v}$. In the following, we define our main objectives with respect to a mechanism $\mathcal{M} = (\mathbf{x}, \mathbf{p})$.

**Definition 5** (Welfare). The welfare obtained by mechanism $\mathcal{M}$ is $W(\mathbf{c}, \mathbf{v}, \mathcal{M}) = \sum_{i=1}^{n} x_i(\mathbf{c}) \cdot (v_i - c_i)$.

**Definition 6** (Utility). The utility obtained by mechanism $\mathcal{M}$ is $U(\mathbf{c}, \mathbf{v}, \mathcal{M}) = \sum_{i=1}^{n} x_i(\mathbf{c}) \cdot v_i - p_i(\mathbf{c})$.

**Approximations.** Since we are bounded by incentive and computational constraints, we devise mechanisms that provide a guaranteed fraction of the optimal solution. Consider some objective function $\mathcal{G}(\mathbf{c}, \mathbf{v}, \mathcal{M})$ (e.g., welfare or utility). Given an allocation function $\mathbf{x}$, consider a (non-truthful) mechanism $\mathcal{M}_{\mathbf{x}}$ that allocates using $\mathbf{x}$, and sets payments exactly as the reported costs, that is $p_i(\mathbf{c}) = x_i(\mathbf{c}) \cdot c_i$. In the prior-free setting, we say that a mechanism $\mathcal{M} = (\mathbf{x}, \mathbf{p})$ $\alpha$-approximates the optimal mechanism for objective $\mathcal{G}$ (for $\alpha \in (0, 1)$), if for every $\mathbf{c}, \mathbf{v}$

$$\mathcal{G}(\mathbf{c}, \mathbf{v}, \mathcal{M}) \geq \alpha \cdot \max_{\text{budget-feasible } \mathcal{M}_{\mathbf{x}}} \mathcal{G}(\mathbf{c}, \mathbf{v}, \mathcal{M}_{\mathbf{x}}).$$

Note that the right-hand side is an overly optimistic benchmark that has full information of the costs of the sellers.

In the Bayesian setting, we say that $\mathcal{M} = (\mathbf{x}, \mathbf{p})$ $\alpha$-approximates the optimal mechanism for objective $\mathcal{G}$ if for

every $\mathbf{v}$ and distribution over costs $\mathcal{D}$,

$$\mathbb{E}_{\mathbf{c} \sim \mathcal{D}}[\mathcal{G}(\mathbf{c}, \mathbf{v}, \mathcal{M})] \geq \alpha \cdot \mathbb{E}_{\mathbf{c} \sim \mathcal{D}}[\max_{\text{budget-feasible } \mathcal{M}_{\mathbf{x}}} \mathcal{G}(\mathbf{c}, \mathbf{v}, \mathcal{M}_{\mathbf{x}})].$$

## 3. Prior-free Procurement for Welfare Maximization and Generalizations

In this section, we present a mechanism that guarantees an approximation to welfare in the prior-free setting. We then show that lower bounds from value maximization extend to the welfare objective, giving a nearly matching lower bound for our mechanism. Finally, we discuss how to adapt the mechanism to handle generalizations of the welfare objective. Our proposed mechanism and analysis closely resemble the mechanisms in (Papadimitriou & Singer, 2010; Chen et al., 2011) for the case of additive value maximization.

The formal description follows, but first, we describe it intuitively. The mechanism initially restricts attention to agents with cost smaller than their value (otherwise, procuring their items only hurts the objective), and smaller than the overall budget $B$ (otherwise, the approximation can be arbitrarily bad). Then, we consider two candidate solutions. The first candidate solution just exclusively procures from the agent with the highest individual welfare that is feasible to serve. The second candidate solution orders agents by decreasing value-per-cost $\frac{v_i}{c_i}$ (or equivalently, welfare-per-cost $\frac{v_i - c_i}{c_i}$) and then greedily includes agents so long as each agent's cost is no higher per unit of value than what the budget can support on average for those added so far. If the welfare-maximizing agent has high enough welfare compared to the optimal fractional solution without it, we procure from this agent, otherwise we procure from agents in the greedy solution. Payments come directly from Myerson's formula.

Our mechanism is formally presented in Mechanism 1. Let $\text{FOPT}(\cdot, \cdot)$ be a function that takes a set of value-cost tuples as the first parameter, and a budget as a second parameter, and computes the optimal greedy fractional solution to the knapsack problem for the parameters given.

Mechanism 1 is a DSIC-IR and budget feasible mechanism that gives a $\frac{1}{2+\sqrt{2}}$-approximation to welfare, and is computable in $O(n \log n)$ time (see Appendix E for details). Due to space limitations, we defer the full proofs to Appendix B, and give the proof sketches of the obtained guarantees.

We first show that the mechanism is DSIC-IR and budget feasible.

**Theorem 2.** *Mechanism 1 is a DSIC-IR and budget feasible mechanism.*

*Proof sketch.* To show the mechanism is DSIC, by Lemma 1, we need to show that the allocation is

---

**Mechanism 1** Welfare-maximizing Budget Feasible Auction

---

1: **Input:** Reported costs $\mathbf{c} = (c_1, \ldots, c_n)$, Known valuations $\mathbf{v} = (v_1, \ldots, v_n)$, Budget $B$
2: **Output:** Allocation set and payments
3: Let $F = \{i : c_i \leq \min(v_i, B)\}$ ▷ Restrict to sellers with positive welfare and feasible cost.
4: Rename bidders in $F$ such that $\frac{v_1}{c_1} \geq \frac{v_2}{c_2} \geq \ldots \geq \frac{v_{|F|}}{c_{|F|}}$.
5: Set $W \leftarrow \emptyset$, $k \leftarrow 1$
6: **while** $k \leq |F|$ and $\frac{c_k}{v_k} \leq \frac{B}{\sum_{j \leq k} v_j}$ **do**
7: $\quad W \leftarrow W \cup \{k\}$
8: $\quad k \leftarrow k + 1$
9: **end while**
10: For every $i \in F$, let $w_i = v_i - c_i$
11: Let $i^* \in \arg\max_{i \in F} w_i$
12: Compute $\text{FOPT}' = \text{FOPT}(\{(w_i, c_i)\}_{i \in F \setminus \{i^*\}}, B)$ ▷ Compute the optimal fractional solution of all agents but $i^*$.
13: **if** $w_{i^*} > \text{FOPT}'/(1 + \sqrt{2})$ **then**
14: $\quad$ Allocate $\{i^*\}$ ▷ Allocate $i^*$ if $i^*$ has high welfare
15: **else**
16: $\quad$ Allocate the set $W$ ▷ Otherwise, allocate $W$
17: **end if**
18: Pay agents using Myersonian payments

---

non-increasing in a seller's reported cost. Whenever some $i \in F$ is allocated, if $i$ is a part of the constructed set $W$, we show that decreasing their reported cost will keep the set $W$ the same, and increases their welfare $w_i$. Therefore, the only change in the allocation might be that $i$ is the welfare-maximizing agent $i^*$ and is the sole seller allocated. If $i \notin W$ and is allocated as the welfare-maximizing seller $i^*$, decreasing their reported cost will only increase $w_{i^*}$ while keeping $\text{FOPT}'$ the same, and therefore $i^*$ is allocated in this case as well.

To show budget-feasibility, we first consider the case $W$ is allocated. In this case, we show that for every $i \in W$, in case they report a cost higher than $T_i = v_i \cdot \frac{B}{\sum_{t \in W} v_t}$, they are no longer in $W$, and no longer allocated. By Observation 1, this implies that when $W$ is allocated, for each $i \in W$, $p_i \leq T_i$ (Lemma 5 in Appendix B), and the total payment is at most $\sum_{i \in W} T_i = \sum_{i \in W} v_i \cdot \frac{B}{\sum_{t \in W} v_t} = B$. When $i^*$ is the sole seller allocated, since we filter-out sellers with cost larger than $B$, budget-feasibility is trivial in this case. $\square$

We next sketch the proof of approximation.

**Theorem 3.** *The welfare obtained by Mechanism 1 is a $\frac{1}{2+\sqrt{2}}$-approximation to the optimal welfare.*

*Proof sketch.* Interestingly, in our mechanism, the sellers were considered to be added to $W$ sorted by the value-over-cost ratio, and added while the value-over-cost of the current seller considered is higher than the total-value-over-budget of all agents considered until now. At first glance it might be surprising that the set $W$ has good welfare guarantees, while it was constructed only using value considerations, where the value from a seller might be arbitrarily larger than the welfare from the same seller. We first observe that after filtering agents with negative welfare, the welfare-over-cost ordering is the same as the value ordering, as

$$w_i/c_i = v_i/c_i - 1 \geq v_j/c_j - 1 = w_j/c_j$$
$$\iff v_i/c_i \geq v_j/c_j.$$

An additional step is needed to prove that the welfare obtained from the sellers in $W$ is a constant approximation to the optimal fractional welfare of all agents minus the welfare of the highest-welfare seller (this is essentially proved through Lemmas 6 and 7).

With this at hand we inspect the two possible solutions: whenever $i^*$ is the sole seller allocated, by the explicit condition of the algorithm, we know the welfare of $i^*$ is close to the welfare of the optimal fractional allocation (and thus, of the optimal integral allocation); otherwise, as $i^*$ cannot contribute too much to the optimal fractional allocation, we can use the welfare guarantees proved for $W$ in order to show that the obtained welfare is near the welfare of the optimal allocation. $\qquad\square$

We complement Mechanism 1's guarantee with a lower bound of $\frac{1}{1+\sqrt{2}}$. We prove this by showing that any lower bound for the (widely-studied) value objective automatically transfers to the welfare objective as well, and thus our lower bound is due to (Chen et al., 2011). The proof of Theorem 4 can be found in Appendix D.1.

**Theorem 4.** *Let $\beta \in (0,1)$. If no budget-feasible mechanism can achieve a better than $\beta$-approximation to the value objective, then no budget-feasible mechanism can achieve a better than $\beta$-approximation to the welfare objective.*

**Corollary 5.** *No deterministic budget-feasible mechanism for welfare maximization can achieve an approximation ratio better than $\frac{1}{1+\sqrt{2}}$.*

In Appendix B.3, we introduce minor adaptations to the mechanism that extend the guarantees to a general family of objective functions, where for each $i$, $w_i = v_i - \alpha \cdot c_i$ for some $\alpha \geq 0$, and we want to maximize $G(\sum_{i \in S} w_i)$, where $G : \mathbb{R}_{\geq 0} \to \mathbb{R}_{\geq 0}$ is a non-decreasing, concave and normalized ($G(0) = 0$) function. Our mechanism follows the footsteps of Mechanism 1, where the filtering step keeps agents where $w_i \geq 0$ and $c_i \leq B$, and FOPT now takes

the new weights $w_i$'s. The proof of DSIC-IR and budget-feasibility remains identical, while the proof of approximation requires subtle modifications. First, we show that the mechanism outputs a set of agents $S$ whose sum of weights $\sum_{i \in S} w_i$ $\beta$-approximates the optimal budget-feasible sum of weights for $\beta = 1/(2 + \sqrt{2})$. To do so, we notice that the weight-over-cost ordering is identical to the value-over-cost ordering, and that the constructed set $W$ has the same approximation guarantees with respect to the optimal sum of weights. We conclude by noticing that since $G$ is monotonically non-decreasing, then the optimal $G(\sum_{i \in S^*} w_i)$ is just the function $G$ applied on the optimal budget-feasible $\sum_{i \in S^*} w_i$, and that

$$G(\sum_{i \in S} w_i) \geq G(\beta \cdot \sum_{i \in S^*} w_i) \geq \beta \cdot G(\sum_{i \in S^*} w_i),$$

where the last inequality follows concavity and normalization of $G$, and $\beta \in (0, 1)$.

The following proposition shows that the buyer's utility objective cannot be approximated when using a prior-free mechanism, motivating switching to the Bayesian setting for this objective. The proof is deferred to Appendix B.5.

**Proposition 6.** *In the prior-free setting, optimal utility is inapproximable even for a single agent.*

## 4. Bayesian Procurement

In this section, we circumvent the impossibility result from Proposition 6 by relaxing the guarantee from worst-case (prior-free) to in expectation (the Bayesian setting). In the Bayesian setting, each seller $i$ has a private cost parameter $c_i$, which is (independently) drawn from a publicly known distribution $F_i$.

We are interested in designing a mechanism $(\mathbf{x}, \mathbf{p})$ that maximizes expected buyer utility, while respecting the budget constraint. We distinguish between ex-post (or *hard*) budget-feasibility and ex-ante (or *soft*) budget-feasibility, as defined in Definition 3 and Definition 4.

Our primary focus is on designing mechanisms that satisfy the hard-budget constraint, since this is the more meaningful requirement: the buyer should never exceed the available budget. To illustrate the distinction, even in the single-seller case, the optimal ex-ante solution might post a price of $B/q > B$, where the seller accepts with probability $q$. In that case, the budget constraint is satisfied only in expectation, while whenever the buyer obtains non-zero utility, the realized payment exceeds the budget. However, we study the soft-budget formulation first en route to an ex post solution: the ex ante problem is analytically tractable and provides a useful relaxation that we leverage to design and analyze approximately utility-optimal mechanisms subject to a hard-budget constraint. Note, however, that in repeated

environments, it may be relevant to study ex-ante feasibility, where temporary budget overruns may average out over time and only long-run expected expenditure must remain bounded.

### 4.1. The Optimal Soft-Budget Mechanism

In this subsection, we provide a solution for the Bayesian soft-budget utility maximization procurement problem. The high-level idea is that because both the objective and the budget constraint are expressed ex ante—in expectation over the agents' costs—the optimal mechanism decomposes across agents: the designer effectively allocates budget to agents in expectation and optimizes each agent's allocation independently. As a result, the global problem reduces to a collection of single-agent optimization problems, whose solutions take the form of posted prices.

The optimization problem at hand is:

$$\max_{x,p} \mathbb{E}_{\mathbf{c}} \left[ \sum_{i \in [n]} (v_i x_i(\mathbf{c}) - p_i(\mathbf{c})) \right]$$

$$\text{s.t. } \mathbb{E}_{\mathbf{c}} \left[ \sum_{i \in [n]} p_i(\mathbf{c}) \right] \leq B, \tag{3}$$

$$(\mathbf{x}, \mathbf{p}) \text{ is DSIC-IR}$$

Previous works (Ensthaler & Giebe, 2014; Balkanski & Hartline, 2016) provide solutions for Bayesian value maximization (i.e., the optimization objective of formulation (3) without the payment term) subject to a soft-budget constraint. The solutions to the value and utility optimization problems end up sharing nearly all of their structural features: under regularity of the virtual cost functions, both admit optimal solutions in the form of independent posted prices. One small conceptual difference is that, in value maximization, it is always optimal to exhaust the entire budget, since allocating additional budget can only increase total value. However, for utility maximization, the soft-budget constraint need not bind: offering an agent too high a price may reduce expected utility, hence a budget surplus may be preferable.

We are able to circumvent this issue using the following observation. If a utility-optimal mechanism spends strictly less than the budget $B$ in expectation, if $B'$ denotes its expected expenditure, the same mechanism is feasible and optimal for the problem with budget $B'$. Consequently, it is without loss to restrict attention to utility maximization in instances in which the budget constraint is tight (we formalize this intuition in the proof of Theorem 7).

Using virtual cost functions, we reformulate the optimization program into a simpler one-variable optimization program, and then characterize the optimal soft-budget mech-

---

**Mechanism 2** Utility-Maximizing Ex-post Budget-Feasible Auction $\mathcal{M}_{\text{const}}(\alpha, \beta; \mathbf{p})$

---

1: Partition the agents into $H = \{i : p_i \geq \frac{B}{\alpha}\}$ and $L = \{i : p_i < \frac{B}{\alpha}\}$, denote their respective ex-ante posted pricing utility as $U_{\text{price}}^H(\mathbf{p}) = \sum_{i \in H} (v_i - p_i) q_i$ and $U_{\text{price}}^L(\mathbf{p}) = \sum_{i \in L} (v_i - p_i) q_i$, and the total ex-ante posted pricing utility as $U_{\text{price}}(\mathbf{p}) = \sum_{i \in L \cup H} (v_i - p_i) q_i$.

2: **if** $U_{\text{price}}^H(\mathbf{p}) \geq \left(1 - \frac{1}{\beta}\right) \cdot U_{\text{price}}(\mathbf{p})$ **then**

3:  Order agents in $H$ by decreasing $v_i - p_i$. Iterate in this order and offer agent $i$ price $p_i$ if the remaining budget is at least $p_i$; otherwise, skip $i$ and continue.

4: **else**

5:  Order agents in $L$ by decreasing $\frac{v_i - p_i}{p_i}$. Iterate in this order and offer agent $i$ price $p_i$ if the remaining budget is at least $p_i$; otherwise, skip $i$ and continue.

6: **end if**

---

anism with the following theorem. The reformulation and the proof are deferred to Appendix C.1.

**Theorem 7.** *Assuming regular cost distributions, there exists a utility-optimal mechanism $\mathcal{M}_{ante}$ for the soft-budget constraint that independently posts prices $\hat{c}_i$ to each agent $i$. Furthermore, these optimal prices $\hat{c}_i$ can be computed efficiently.*

### 4.2. A Constant Approximation Subject to a Hard Budget Constraint

In this section, we construct a mechanism that gives a constant-factor approximation to the *ex-post* budget-feasible optimal utility. Notice that enforcing budget feasibility ex-post is only more restrictive than enforcing the budget only in expectation, since every ex-post budget-feasible mechanism is also feasible ex-ante, while the converse need not hold. Consequently, the optimal utility subject to *ex-ante* budget-feasibility upper bounds the optimal utility subject to ex-post budget-feasibility, that is $\text{OPT}_{ante} \geq \text{OPT}_{post}$.

At a high level, our mechanism will take as input a price vector $\mathbf{p}$ that is ex-ante budget feasible and satisfies $p_i \leq B$ for every agent. For these prices, define the ex-ante posted pricing utility benchmark $U_{\text{price}}(\mathbf{p}) = \sum (v_i - p_i) q_i$.[1] Using $\mathbf{p}$, we partition agents into high-price agents $H$ and low-price agents $L$: if $U_{\text{price}}(\mathbf{p})$ derives most of its utility from $H$, we run a sequential posted-price procedure on $H$ in decreasing order of utility $v_i - p_i$; otherwise, we run a sequential posted-price procedure on $L$ in decreasing order of utility-per-payment $(v_i - p_i)/p_i$. The formal description appears in Mechanism 2.

---

[1] $q_i = q_i(p_i) = F_i(p_i)$, but for ease of notation, we omit dependence on $p$.

The ex-ante prices from Section 4.1 may not be suitable for our ex-post mechanism: while they always satisfy $\hat{c}_i \leq v_i$, they may individually exceed the budget (i.e., $\hat{c}_i > B$). We therefore instead use the prices obtained by solving the soft-budget problem on the truncated instance $F^{\leq B}$, where costs above $B$ are capped at $B$.

This solution may no longer be optimal for the original soft-budget problem, but preserves the posted-price structure under regularity, guarantees $\hat{c}_i \leq B$, and remains a valid relaxation because any ex-post budget-feasible mechanism never offers prices above $B$. Hence:

$$\text{OPT}_{ante}(F^{\leq B}) \geq \text{OPT}_{post}(F^{\leq B}) = \text{OPT}_{post}. \quad (4)$$

Our analysis distinguishes between two cases. When low-priced agents contribute most of the utility, we adapt the analysis of Balkanski & Hartline (2016), that addresses the Bayesian value-maximization problem to the utility objective. Importantly, we identify and correct a subtle but important error in the analysis of one of their lemmas (Remark 1). This issue does not undermine the validity of their main theorem; rather, we revisit the argument and supply a modified proof for clarity, correctness, and completeness. The approach of (Balkanski & Hartline, 2016) holds whenever the maximal price is not too high, and does not give a constant factor approximation for high-priced agents. When the high-priced agents contribute most of the utility to the ex-ante objective, we show that a posted price mechanism that orders agents according to their utility extracts a constant factor of this value.

We first address the case where low-priced agents contribute the most to the ex-ante objective. As in Balkanski & Hartline (2016), let $k = \min_i \frac{B}{p_i}$. The smaller $k$ is, the higher the maximal price is.[2] Consider $\mathcal{M}(\boldsymbol{p})$, a sequential posted pricing mechanism that orders agents by decreasing $\frac{v_i - p_i}{p_i}$ and offers them prices $p_i$, as described in Line 5 of Mechanism 2. The next lemma analyzes the approximation ratio of this mechanism (proven in Appendix C.2).

**Lemma 2.** *Assuming prices $\boldsymbol{p}$ that satisfy the budget constraint ex-ante and are individually ex-post feasible ($p_i \leq B$), $\mathcal{M}(\boldsymbol{p})$ achieves a $(1 - \frac{1}{\sqrt{2\pi k}})(1 - \frac{1}{k})$-approximation to the ex-ante posted pricing utility $U_{price}(\boldsymbol{p})$.*

We now provide our approximation guarantee and analysis of Mechanism 2, and afterward, in Corollary 9, we optimize these values to show that our guarantee is in fact at

least a $1/5$-approximation. We note that Mechanism 2 is computable in $O(n \log n)$ time (see Appendix E).

**Theorem 8.** *Assuming prices $\boldsymbol{p}$ that satisfy the budget constraint ex-ante and are individually ex-post feasible ($p_i \leq B$), for any parameters $\alpha > 1$ and $\beta > 1$, the utility-maximizing ex-post budget-feasible auction $\mathcal{M}_{const}(\alpha, \beta; \boldsymbol{p})$ approximates the ex-ante posted pricing utility $U_{price}(\boldsymbol{p})$ with an approximation ratio of*

$$\min\left(\frac{1}{\beta}\left(1 - \frac{1}{\alpha}\right)\left(1 - \frac{1}{\sqrt{2\pi\alpha}}\right), \left(1 - \frac{1}{\beta}\right)\frac{1 - e^{-\alpha}}{\alpha}\right).$$

*Proof.* Let the expected utility of mechanism $\mathcal{M}_{const}$[3] be $U_{const}(\boldsymbol{p})$. We consider the two cases of the mechanism $\mathcal{M}_{const}$.

**Case 1:** The ex-ante posted pricing utility of the high agents $H$ is a good approximation to the total ex-ante posted pricing utility, that is, $U_{price}^H(\boldsymbol{p}) \geq \left(1 - \frac{1}{\beta}\right) \cdot U_{price}(\boldsymbol{p})$. We will lower bound the performance of mechanism $\mathcal{M}_{const}$ using the following lemma (the proof of which is deferred to the Appendix C.2).

**Lemma 3.** *Assuming prices $\boldsymbol{p}$ that satisfy the budget constraint ex-ante, are individually ex-post feasible ($p_i \leq B$) and additionally satisfy $p_i \geq \frac{B}{\alpha}$, the sequential posted pricing mechanism that orders agents by decreasing $v_i - p_i$ and offers them prices $p_i$, achieves a $\frac{1 - e^{-\alpha}}{\alpha}$-approximation to the ex-ante posted pricing utility $U_{price}(\boldsymbol{p})$.*

The assumptions of the lemma are met by agents in $H$ and prices $\boldsymbol{p}$, and as such, combining the lemma with the case assumption, we get:

$$U_{const}(\boldsymbol{p}) \geq \frac{1 - e^{-\alpha}}{\alpha} \cdot \left(1 - \frac{1}{\beta}\right) \cdot U_{price}(\boldsymbol{p}) \quad (5)$$

**Case 2:** The ex-ante posted pricing utility of the low agents $L$ is a good approximation to the total ex-ante posted pricing utility, that is, $U_{price}^L(\boldsymbol{p}) \geq \frac{U_{price}(\boldsymbol{p})}{\beta}$. By definition of the set $L$, the posted prices used by the mechanism (in this subcase) satisfy $p_i \leq \frac{B}{\alpha}$. Then in this case, Lemma 2 with $k = \alpha$:

$$U_{const}(\boldsymbol{p}) \geq \frac{1}{\beta}(1 - \frac{1}{\sqrt{2\pi\alpha}})(1 - \frac{1}{\alpha})U_{price}(\boldsymbol{p}). \quad (6)$$

In conclusion, the expected utility of mechanism $\mathcal{M}_{const}$ is always at least the minimum of the approximation guarantees of the two cases, Equations (5) and (6), so we get:

$$\frac{U_{const}(\boldsymbol{p})}{U_{price}(\boldsymbol{p})} \geq \min\{\frac{1}{\beta}(1 - \frac{1}{\alpha})(1 - \frac{1}{\sqrt{2\pi\alpha}}),$$
$$(1 - \frac{1}{\beta})(\frac{1 - e^{-\alpha}}{\alpha})\}. \quad (7)$$

---

[2] In fact, the domain of parameter $k$ is not explicitly addressed in (Balkanski & Hartline, 2016): their approximation guarantee is meaningful only when $k \geq 1$, while the methodology used to generate prices may induce instances with $k < 1$, for which no guarantee is provided, and the mechanism may return an empty outcome despite feasibility. In contrast, our approach applies uniformly regardless of $k$.

[3] For ease of notation, we omit the dependence $\alpha, \beta$.

Finally, for the special case where $\boldsymbol{p} = \hat{\boldsymbol{c}}$ (where $\hat{c}$ are the prices of the optimal solution of the ex-ante problem on the truncated distributions $F^{\leq B}$) we note that $U_{\text{price}}(\boldsymbol{p}) = U_{\text{price}}(\hat{\boldsymbol{c}}) = \text{OPT}_{ante}(F^{\leq B})$ and by Equations (4) and (7) we get:

$$\frac{U_{\text{const}}(\hat{\boldsymbol{c}})}{\text{OPT}_{post}} \geq \min\{\frac{1}{\beta}(1 - \frac{1}{\alpha})(1 - \frac{1}{\sqrt{2\pi\alpha}}),$$
$$(1 - \frac{1}{\beta})(\frac{1 - e^{-\alpha}}{\alpha})\}.$$

□

The above theorem holds for regular cost distributions and with $\hat{c}$ being the optimal prices of the ex-ante problem on the truncated distributions $F^{\leq B}$ as prices $\boldsymbol{p}$. We get the following corollary.

**Corollary 9.** *The approximation guarantee in Theorem 8 is maximized by $\alpha_0 \approx 2.41$ and $\beta_0 \approx 2.15$. For these parameters, the mechanism $\mathcal{M}_{const}(\alpha_0, \beta_0; \hat{\boldsymbol{c}})$ achieves a constant approximation ratio of at least $0.2021$ to the utility-optimal ex-post budget-feasible $\text{OPT}_{post}$.*

The corollary follows by equating the two expressions in the minimum of Theorem 8, substituting for $\beta = \beta(\alpha)$ and optimizing numerically over $\alpha$. Finally, we note that the standalone mechanism's $\mathcal{M}(\boldsymbol{p})$ approximation implied by Lemma 2 becomes better than the one proved in Theorem 8 for Mechanism 2 whenever $k \geq 1.43$.

Additionally, we highlight the components of our analysis that are tight, pertaining to the conversion of ex-ante posted prices into sequential posted-price mechanisms that are ex-post feasible. Specifically, we show in Appendix D.2 that the $1 - \frac{1}{e}$ factor in Lemma 3 is tight (see Lemma 11), while the $1 - O(1/\sqrt{k})$ approximation guarantee in Lemma 2 is tight up to constants (see Lemma 12). Both results are proven using simple symmetric instances in which posted prices are identical across agents, and hence the expected utility of any sequential posted-price mechanism depends only on the set of accepting agents, rather than on the ordering rule.

### 4.3. Generalized Objective - Irregularity

In this subsection, we relax the regularity assumption of Sections 4.1 and 4.2 and we generalize the analysis and results to a larger class of objectives.

#### 4.3.1. IRREGULAR DISTRIBUTIONS

**Soft-budget Optimal Mechanism (Irregular Distributions).** When cost distributions are irregular, we apply the standard ironing reduction Myerson (1981) (as in Balkanski & Hartline, 2016 for welfare). Concretely, we define an auxiliary optimization program obtained by replacing each

virtual-cost function with its *ironed* virtual-cost function, which satisfies regularity. The key step is to show that an optimal solution to this ironed program can be converted into a solution for the original program while preserving feasibility and the achieved objective value; hence the two programs have the same optimum. We defer the definition of the ironed program and the ironing-to-original conversion argument to Appendix C.3. The resulting optimal mechanism is characterized by the following theorem.

**Theorem 10.** *There exists a utility-optimal mechanism $\mathcal{M}_{ante}$ for the soft-budget problem that posts prices independently across agents. For each agent $i$, $\mathcal{M}_{ante}$ either (i) posts a single price $\hat{c}_i$, or (ii) randomizes between two prices by posting $\hat{c}_{i,1}$ with probability $\theta_i \in [0, 1]$ and $\hat{c}_{i,2}$ with probability $1 - \theta_i$. Moreover, the corresponding optimal prices and mixing probabilities $(\hat{\boldsymbol{c}}, \boldsymbol{\theta})$ can be computed efficiently.*

**Hard-budget Approximately Optimal Mechanism.** Mechanism 2 requires a *single* posted price for each agent. Under irregular cost distributions, however, the optimal ex-ante mechanism from Theorem 10 may randomize for some agent $i$ between two prices. To reconcile this with Mechanism 2, we define the sets $H$ and $L$ over all candidate prices rather than over agents, and modify $U^H_{\text{price}}$ and $U^L_{\text{price}}$ by weighting each term with the probability with which each candidate price is used. The mechanism will then use the modified $U^H_{\text{price}}, U^L_{\text{price}}$ to decide which branch of Mechanism 2 to execute; it then realizes one price for each agent and runs the selected branch only on the realized agents whose sampled price belongs to that branch. We denote this mechanism by $\mathcal{M}^{\text{rand}}_{\text{const}}$ (definition in Mechanism 4), and prove the following corollary:

**Corollary 11.** *Let $(\hat{\boldsymbol{c}}, \boldsymbol{\theta})$ be an optimal solution of Theorem 10 on the truncated distributions. The utility-maximizing ex-post budget-feasible auction $\mathcal{M}^{\text{rand}}_{const}(\alpha_0, \beta_0; \hat{\boldsymbol{c}}, \boldsymbol{\theta})$ achieves a constant approximation ratio of at least $0.2021$ to the utility-optimal ex-post budget-feasible $\text{OPT}_{post}$.*

*Proof sketch.* To prove this result we need to consider the two cases of the mechanism. In the high-price case, where the mechanism orders by utility (line 4), all arguments in the proof of Lemma 3 go through if we write $q_i$ and $p_i$ in terms of $\theta_i$. In the low-price case, where the mechanism orders by utility over price (line 6), we can apply Lemma 2 to a deterministic instance: each agent with a randomized price is replaced by one copy for each candidate price, with the same value and appropriately weighted (by $\theta_i$) acceptance probability. This preserves ex-ante utility and expected payments, but removes the mutual exclusivity among price realizations, thereby (potentially) increasing ex-post budget contention. We can then prove that the deterministic

instance has weakly lower expected budget-feasible utility than the original randomized instance (see (Balkanski & Hartline, 2016, Theorem 32) for the full argument). □

### 4.3.2. DIFFERENT OBJECTIVES

**Generalized Utility Objective.** The mechanisms and guarantees in Sections 4.1 and 4.2 extend to any additively separable affine objective with nonnegative weights on values and payments: for coefficients $\{a_i, b_i\}_{i \in [n]}$, maximize the *generalized utility*

$$\mathbb{E}_{\mathbf{c}}\left[\sum_i (a_i v_i x_i(\mathbf{c}) - b_i p_i(\mathbf{c}))\right]$$

under the same DSIC-IR and budget constraints.

Under regularity, the optimal soft-budget mechanism remains independent posted pricing (and under irregularity, possibly randomizing over two prices for each agent); the hard-budget reduction applies verbatim by replacing per-agent utility with $g_i = a_i v_i - b_i \hat{c}_i$, yielding the same constant-factor approximation (details in Appendix C.3).

**Ironing limitations.** Ironing worked above because both the objective and the (ex-ante) budget constraint depend on the interim rule through the same linear function $\int \phi_i(q) x_i(q) \, dq$ (and thus $\int (v_i - \phi_i(q)) x_i(q) \, dq$), so averaging on ironed intervals preserved value. If the objective is not linear in $\phi_i(\cdot) x_i(\cdot)$, then Lemma 10 need not be tight for the objective and constraint simultaneously: flattening to preserve feasibility can strictly decrease the objective, while averaging to preserve the objective can violate the budget constraint. This suggests that our generalized utility family may capture the natural frontier of objectives for which ex-ante optimality via ironing continues to hold.

**Other objectives.** Regularity in Section 4.1 was used to ensure that each single-agent Lagrangian subproblem is optimized by a *single* posted price; more generally, this conclusion continues to hold for any objective that is additive across agents and (weakly) decreasing in cost, so the resulting ex-ante optimal mechanism remains independent posted pricing (and can again be used as input to our hard-budget reduction to obtain an approximately optimal ex-post mechanism). Additionally our ex-post approximation framework does not require exact optimality of the ex-ante solution: for any additive objective admitting independent posted prices that are ex-ante budget feasible, individually ex-post feasible (i.e., each price is at most $B$), and achieve an $\alpha$-approximation to the corresponding ex-ante truncated benchmark, these prices can be used as input to $\mathcal{M}_{\text{const}}$ to obtain a constant-factor approximation to the optimal ex-post value.

## 5. Discussion

In this work we extend the literature on budget-feasible procurement to more general objectives. In the prior-free setting, we provide a class of extremely simple, efficient mechanisms with constant-factor guarantees for welfare (or generalizations where seller costs are weighted by $\alpha \geq 0$). In the Bayesian setting, which is necessary for the objective of buyer utility, we also provide a class of extremely simple, efficient mechanisms with constant-factor guarantees, this time for any objective where value and payment are weighted by $a_i, b_i \geq 0$ respectively.

There are many interesting directions for future work. One interesting direction may be to split the gap between the prior-free and Bayesian settings by examining prior-independent mechanism design or learning from samples: both (1) how many samples are required to effectively learn the sellers' cost distributions to approximately implement this optimization and (2) whether one can effectively use samples to run a mechanism without learning the distributions first. Both of these directions have been tackled for other areas of mechanism design (e.g., revenue maximization, see for example the introduction of (Guo et al., 2019)). Armed with the solution of the Bayesian problem to use as a benchmark, one can now investigate sample complexity questions and best understand how sample-complexity bounds trade off with approximations and fit into the bigger picture.

Another interesting direction is to study simple mechanisms for general objectives beyond the additive (knapsack) setting. While additive valuations are frequently seen in practice, there are also some scenarios where they do not capture a buyer's value, i.e., where the marginal value of procured goods might depend on other procured goods. In such cases, the mechanisms will need to be adapted. However, we believe that basic ideas used in our mechanisms, such as (marginal) value-over-cost ordering and pricing based on soft budget relaxation, will be useful in such settings as well.

One other interesting question is to consider production constraints on the part of the sellers. For instance, in the example of API access to language models, in some cases, specialized LLMs might have limited compute, which might introduce an additional layer of competition among buyers, and which buyers need to take into account when setting prices. However, the way to model this would be in a multi-buyer setting, which in addition to a procurement setting, becomes a two-sided market with budgeted buyers. This would be a very interesting problem for future work, introducing all of the additional complexities that come with two-sided markets.

## Impact Statement

This paper presents work whose goal is to advance the field of Machine Learning. There are many potential societal consequences of our work, none which we feel must be specifically highlighted here.

## Acknowledgements

The work of A. Eden and E. Kerner was supported by the Israel Science Foundation (grant No. 533/23). Alon Eden is the Incumbent of the Harry & Abe Sherman Senior Lectureship at the School of Computer Science and Engineering at the Hebrew University. The work of K. Goldner and T. Tsilivis was supported by NSF CAREER Award CCF-2441071.

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

# A. Additional Related Work

Our work is situated at the intersection of budget-feasible mechanism design and optimal auction theory. In this section, we review the development of budget-feasible mechanisms, distinguishing between works in the prior-free setting, the Bayesian setting, and more general settings whose approaches we cannot use. We refer the reader to Liu et al. (2024) for a comprehensive survey of budget-feasible procurement mechanisms. We also review what is known for Bayesian utility maximization.

**Value Maximization in Prior-Free Budget-Feasible Procurement.** The prior-free budget-feasible procurement problem was first introduced by Singer (2010), who observed that classic techniques (such as the VCG mechanism) cannot be used as they may violate the budget constraint. Singer (2010) aims to maximize the buyer's value subject to budget-feasibility, and for submodular valuations, develops randomized mechanisms that achieve constant-factor approximations to the optimal value. Notably, an early preprint (Papadimitriou & Singer, 2010) includes a much simpler mechanism for the case of additive ("knapsack") values.

Follow-up work by Chen et al. (2011) gives deterministic and randomized mechanisms with improved approximation ratio for submodular functions. For additive values, they refine the techniques introduced in (Papadimitriou & Singer, 2010) and provide a $1/(2 + \sqrt{2})$-approximate deterministic mechanism and a $\frac{1}{3}$-approximate randomized mechanism. They also provide impossibilities of $1/(1 + \sqrt{2})$ and $1/2$ for deterministic and randomized mechanisms, respectively. Gravin et al. (2020) further improve the approximation ratios for additive valuations, provide an optimal $\frac{1}{2}$-approximation randomized mechanism, and an improved $\frac{1}{3}$-approximation deterministic mechanism.

Anari et al. (2018) and Rubinstein & Zhao (2023) study value maximization in the prior-free setting subject to a hard budget constraint under a large market assumption and provide tight $(1 - \frac{1}{e})$-approximate mechanisms for additive valuations. Anari et al. (2018) additionally provides constant approximation mechanisms for submodular valuation functions.

Finally, concurrently and independently of our work, Cui et al. (2026) study budget-feasible mechanisms for submodular welfare maximization in procurement auctions. They provide mechanisms that guarantee constant-factor approximations subject to a small additive error for settings with both monotone and non-monotone submodular welfare.

**Bayesian Budget-Feasible Procurement.** As we show in Proposition 6, for the budget-feasible procurement problem, obtaining any approximation to the optimal buyer utility in the prior-free setting is hopeless. As a result, we turn to the Bayesian setting. For the setting without a budget, Myerson (1981) gives a complete characterization of the optimal auction, as well as unique payments that satisfy incentive compatibility and truthfulness.

The budget-feasible procurement problem has been studied in the Bayesian setting before, but primarily for the objective of value maximization. The first work in this space is by Ensthaler & Giebe (2014), who solve the Bayesian value-maximizing problem subject to a soft budget constraint (that need only hold in expectation over the realization of seller costs); their work also assumes that seller cost distributions satisfy regularity. Balkanski & Hartline (2016) extend this work for irregular distributions, and use it to address the problem with a hard budget constraint, that is, when the budget must be satisfied for every realization of the sellers' costs (ex-post). They construct a simple posted pricing mechanism that achieves a $\left(1 - O(\frac{1}{\sqrt{k}})\right)$-approximation when all optimal ex-ante prices take up at most $1/k$-th of the budget $B$ (see Section 4.2 for details).

Most related to our work is Jarman & Meisner (2017) that studies the ex-post budget-feasible procurement problem for the objective of utility maximization. Importantly, they restrict themselves only to deterministic mechanisms. They devise properties of the optimal mechanism (characterizing it fully for symmetric agents and partially for asymmetric), and detail how they can implement said mechanism using descending clock auctions. They additionally remark on how their ideas extend to objectives that are convex combinations of values minus payments, similarly to our work. We emphasize that our results approximate the randomized (unrestricted) utility-optimal mechanism subject to ex-post budget-feasibility, which is a harder benchmark to understand and approximate.

Bei et al. (2012) study ex-post budget-feasible *combinatorial* procurement with subadditive and XOS valuation functions, in both the prior-free and Bayesian settings. In the prior-free model, they give mechanisms achieving constant-factor approximations for XOS valuations and sublogarithmic approximations for subadditive valuations. In the Bayesian model, they give a constant-factor approximation for subadditive valuations.

Most recently, (Bhawalkar et al., 2025b) studied a Bayesian value-maximization procurement setting in which sellers can offer multiple services. They develop a general framework that uses (modified) online contention resolution schemes (OCRS) to convert interim allocations and prices into an ex-post feasible procurement mechanism. Instantiating this framework with the OCRS of (Jiang et al., 2025) yields a 0.319-approximation, while their own newly developed OCRS gives a 1/6-approximation overall, together with improved guarantees on certain subinstances. Their setting is inherently more challenging than ours, since sellers are multidimensional rather than single-dimensional. However, the resulting OCRS-based mechanisms are also considerably more complicated and harder to implement in practice. Moreover, their mechanisms satisfy the weaker truthfulness notion of Bayesian Incentive Compatibility (BIC), whereas our mechanisms satisfy the strongest standard notion, namely Dominant-Strategy Incentive Compatibility (DSIC).

**Other Input Models.**   Charalampopoulos et al. (2025) study prior-free, random-order online budget-feasible procurement, giving a randomized posted-price mechanism with a constant competitive ratio for monotone submodular valuations (with a large constant even for additive valuations). Amanatidis et al. (2025) augment this setting with predictions, designing truthful, budget-feasible mechanisms with improved competitive ratios for monotone and non-monotone submodular objectives given a prediction of offline OPT.

**Utility Maximization.**   One of our primary objectives in this paper is maximizing the buyer's utility: value minus payments to the sellers. This objective, known by many names (buyer utility, consumer surplus, residual surplus, money burning, settings with ordeals) was first explored in the algorithmic community by Hartline & Roughgarden (2008). They study a forward auction, where the mechanism designer is a single seller determining how to sell $k$ identical goods to unit-demand buyers. In the Bayesian setting, they give an Myersonian-like characterization of the optimal mechanism, and they also provide a prior-free mechanism that gives a constant-approximation to the prior-free benchmark they offer and a tight $\Theta(1 + \log \frac{n}{m})$-approximation to social welfare. Qiao & Valiant (2023) consider an online pen testing in the random order setting, which is equivalent to using posted price mechanisms. They provide an $O(1/\log n)$-approximation to the optimal social welfare. Goldner et al. (2026) consider the random-order single-item utility maximization problem, but augmented with predictions. They provide mechanisms that achieve a constant-factor approximation to the optimal social welfare when the predictions are accurate (consistency) and a constant-factor approximation to the optimal utility when the predictions are arbitrarily bad (robustness).

Additional work has studied Bayesian utility maximization in increasingly complex environments—service feasibility constraints (Ganesh & Hartline, 2025), i.i.d. heterogeneous items (Goldner & Lundy, 2025), general unit-demand (Ezra et al., 2025), or general outcomes (Fotakis et al., 2016)—providing approximation guarantees to the upper bound of social welfare. Notably, all of these works are for the setting of forward auctions (1 seller, $n$ buyers). One relevant work focuses procurement auctions (1 buyer, $n$ sellers), but does not have a budget constraint, and is focused on equilibrium analysis rather than mechanism design Essaidi et al. (2024). In contrast, we focus on mechanism design in the procurement setting (1 buyer, $n$ sellers) with a budget constraint.

## B. Missing proofs of Section 3

### B.1. Proof of DSIC-IR and Budget-Feasibility

For every $i \in W$, we define $T_i = v_i \cdot \frac{B}{\sum_{t \in W} v_t}$. The values of $T_i$ serve as an upper bound on the payment of the sellers in $W$ in the case that $W$ is allocated, ensuring budget-feasibility. We first show that every seller who reports a cost greater than $T_i$ will not be included in $W$.

**Lemma 4.** *Consider the case where $i \in W$ reports $\hat{c}_i > T_i$, and all other sellers' reports are fixed. Let $\hat{W}$ be the newly constructed set $W$, when Mechanism 1 receives the new reports. Then $W_{-i} \subseteq \hat{W}$ and $i \notin \hat{W}$.*

*Proof.* Consider $i$ changing its reported cost to $\hat{c}_i > T_i$, while all other agents fix their reports. If $\hat{c}_i > \min(v_i, B)$, then $i \notin F$, and clearly $i \notin W$. Otherwise, we first show that $i$ is after all agents in $W_{-i}$ in the value-over-cost order. Denote by $k$ the last item inserted in $W$ when agents report **c**. We know that for every $j \in W_{-i}$, we have that

$$c_j/v_j \ \leq \ c_k/v_k \ \leq \ \frac{B}{\sum_{t \in W} v_t} \ = \ T_i/v_i \ < \ \hat{c}_i/v_i,$$

where the first inequality follows the greedy ordering, the second inequality follows from the fact that $k$ is added to $W$ when

bidders report $\mathbf{c}$, and that $k$ is considered after $W_{-k}$ is added to $W$, the equality follows the definition of $T_i$, and the last inequality follows from $\hat{c}_i > T_i$.

We notice that $W_{-i} \subseteq \hat{W}$. Consider some $j \in W_{-i}$; notice that until $j$ is considered, either the exact same set of agents is considered to be added to $\hat{W}$, or the same set without $i$. Therefore, the condition they face can only be made less strict, and the algorithm adds them to $\hat{W}$ as well.

We finally claim that $i$ will not be added to $\hat{W}$ for reports $(\hat{c}_i, \mathbf{c}_{-i})$. Either the algorithm does not consider $i$ because some earlier agent in the value-over-cost order does not satisfy the condition to be added to $\hat{W}$; otherwise, by the above, when $i$ is considered, the constructed set $\hat{W}$ already contains agents in $W_{-i}$, and we have that

$$\hat{c}_i/v_i > T_i/v_i = B/\sum_{t \in W} v_t \geq B/\sum_{t \in \hat{W} \cup \{i\}} v_t.$$

Therefore, $i$ is not added to $\hat{W}$. □

The following lemma establishes that $T_i$ indeed bounds the payment of agent $i \in W$ in case $W$ is allocated.

**Lemma 5.** *For every $i \in W$, if $W$ is allocated, then $p_i \leq T_i$.*

*Proof.* We show that for every $i \in W$, whenever they report $\hat{c}_i > T_i$, they are not allocated. By Lemma 4, whenever $i \in W$ reports $\hat{c}_i > T_i$, they are excluded from $W$. For agents $i \in W \setminus \{i^*\}$, it is clear that reporting a higher cost will keep them unallocated since they would not become $i^*$. For $i^* \in W$, since $W$ is allocated, we know that

$$w_{i^*} = v_{i^*} - c_{i^*} \leq \text{FOPT}'/(1 + \sqrt{2}). \tag{8}$$

If $i^*$ reports a higher cost, then $w_{i^*} = v_{i^*} - c_{i^*}$ decreases, and $\text{FOPT}'$ stays the same. Since $i^* \notin W$, $i^*$ is not allocated when reporting $\hat{c}_i > T_i$.

As no agent in $W$ is allocated when reporting $\hat{c}_i > T_i$, by Observation 1, it follows that $p_i \leq T_i$ for every such agent. □

**Observation 12.** *If $i \in W$ when reporting $c_i$, then the same set $W$ is constructed when reporting $\hat{c}_i < c_i$.*

*Proof.* $i$ is no later in the value-over-cost ordering when reporting $\hat{c}_i$, and $\hat{c}_i/v_i < c_i/v_i$. Therefore, if the condition to be included in $W$ is met when reporting $c_i$, it is met when reporting $\hat{c}_i$ as well. Now consider $j < i$. Obviously, $j \in W$ when $i$ reports $c_i$. If $j$ is considered before $i$ in the greedy ordering when $i$ reports $\hat{c}_i$, then $j$ is still added to $W$. If $j$ is considered after $i$, then

$$c_j/v_j \leq c_i/v_i \leq B/\sum_{t \leq i} v_t \leq B/\sum_{\{t \leq j\} \cup \{i\}} v_t,$$

and $j$ is still added to $W$. Notice that if $j > i$, then the same agents will be added to $W$ until $j$'s turn when $i$ reports $\hat{c}_i$, and therefore, $j$ will be added to $W$ in this case as well. Whenever the first $j \in F \setminus W$ is considered, all the agents of $W$ are already added, so $j$ would not satisfy the condition to be added to $W$, as before. □

**Theorem 2.** *Mechanism 1 is a DSIC-IR and budget feasible mechanism.*

*Proof.* We first show that each seller's allocation is monotonically non-increasing in their reported cost, which implies DSIC-IR, as we use Myersonian payments. Consider an agent that is allocated when reporting $c_i$. We show that for every report $\hat{c}_i < c_i$, $i$ is still allocated. Consider the case $i$ is $i^*$, the agent maximizing $w_i = v_i - c_i$. If $i$ lowers their cost to $\hat{c}_i$, it remains $i^*$. If $i^*$ is in $W$ when reporting $\hat{c}_{i^*}$, then it is allocated in any case. If $i^*$ is not in $W$ when reporting $\hat{c}_{i^*}$, by Observation 12, $i^*$ is not in $W$ when reporting $c_{i^*}$ as well. Thus, $v_{i^*} - \hat{c}_{i^*} > v_{i^*} - c_{i^*} > \text{FOPT}'/(1 + \sqrt{2})$, implying that $i^*$ is still selected. Now consider $i$ which is different than $i^*$ for report $c_i$, and $i \in W$. For a lower report $\hat{c}_i < c_i$, by Observation 12, the same set $W$ is constructed for report $\hat{c}_i$. Therefore, if $i$ becomes $i^*$ when reporting $\hat{c}_i$, it is allocated in any case. If $i$ does not become $i^*$, since $i^*$ was not selected before, it is not selected in this case as well, and $i$ is allocated. We get that the allocation is monotone, and by Myerson's Lemma, DSIC-IR follows.

It is only left to show budget-feasibility. Assume that $W$ is allocated. Then, by Lemma 5, since every $i \in W$ is not allocated for $\hat{c}_i > T_i$, then $p_i \leq T_i$. Thus, the total payment of all allocated bidders is at most

$$\sum_{i \in W} T_i = \sum_{i \in W} v_i \cdot \frac{B}{\sum_{j \in W} v_j} = B.$$

Whenever $W$ is not allocated, and $i^*$ is allocated alone, since every agent with $c_i > B$ is not in $F$, then $p_{i^*} \leq B$. In both cases, the mechanism is budget feasible. $\qquad\square$

## B.2. Proof of Approximation

Let $k$ be the index of the last agent inserted to $W$ and $\ell$ the largest index such that $\sum_{i \leq \ell} c_i \leq B$. If $\ell < |F|$, let $c'_{\ell+1} = B - \sum_{i \leq \ell} c_i$, $v'_{\ell+1} = v_{\ell+1} \cdot \frac{c'_{\ell+1}}{c_{\ell+1}}$ and $w'_{\ell+1} = v'_{\ell+1} - c'_{\ell+1} = w_{\ell+1} \cdot \frac{c'_{\ell+1}}{c_{\ell+1}}$. Note that the welfare obtained by the optimal fractional greedy allocation is exactly $\sum_{i \leq \ell} w_i + \mathbb{I}_{\ell < |F|} \cdot w'_{\ell+1}$.

The following lemma formalizes the intuition that the welfare-over-cost ratio of the tail of the agents that we do not add to $W$ with index $\leq \ell$ is no better than the welfare-over-cost ratio of the first agent we do not take.

**Lemma 6.** *When $\ell > k$,*

$$\frac{\sum_{i=k+1}^{\ell} w_i + \mathbb{I}_{\ell < |F|} \cdot w'_{\ell+1}}{\sum_{i=k+1}^{\ell} c_i + \mathbb{I}_{\ell < |F|} \cdot c'_{\ell+1}} \leq \frac{w_{k+1}}{c_{k+1}}.$$

*Proof.* Notice for every agent $i$, $w_i/c_i = v_i/c_i - 1$. Thus, for every $i, j$ $w_i/c_i \leq w_j/c_j \iff v_i/c_i \leq v_j/c_j$. This means that sorting the agents according to value-over-cost ratio, also sorts them according to welfare-over-cost ratio, implying that for every $i \in [k+1, \ell]$,

$$w_i/c_i \leq w_{k+1}/c_{k+1} \Rightarrow w_i \leq c_i \cdot w_{k+1}/c_{k+1}.$$

Whenever $\ell < |F|$, we also have that $w'_{\ell+1}/c'_{\ell+1} = w_{\ell+1}/c_{\ell+1} \leq w_{k+1}/c_{k+1}$, implying that $w'_{\ell+1} \leq c'_{\ell+1} \cdot w_{k+1}/c_{k+1}$ as well. Summing over all such $i$'s, we get

$$\sum_{i=k+1}^{\ell} w_i + \mathbb{I}_{\ell < |F|} \cdot w'_{\ell+1} \leq \frac{w_{k+1}}{c_{k+1}} \cdot \left( \sum_{i=k+1}^{\ell} c_i + \mathbb{I}_{\ell < |F|} \cdot c'_{\ell+1} \right),$$

as desired. $\qquad\square$

The following lemma shows that with the welfare of the first item not added to $W$ by the mechanism, the obtained welfare is a good approximation of the optimal welfare.

**Lemma 7.** *When $\ell > k$,*

$$\sum_{i \leq k+1} w_i \geq \sum_{k+1 \leq i \leq \ell} w_i + \mathbb{I}_{\ell < |F|} \cdot w'_{\ell+1}.$$

*Proof.* Assume towards contradiction that

$$\sum_{i \leq k+1} w_i < \sum_{k+1 \leq i \leq \ell} w_i + \mathbb{I}_{\ell < |F|} \cdot w'_{\ell+1}. \tag{9}$$

We have that

$$
\begin{aligned}
v_{k+1}/c_{k+1} &= \frac{v_{k+1} - c_{k+1}}{c_{k+1}} + 1 = \frac{w_{k+1}}{c_{k+1}} + 1 \\
&\underset{(10)}{\geq} \frac{\sum_{i=k+1}^{\ell} w_i + \mathbb{I}_{\ell < |F|} \cdot w'_{\ell+1}}{\sum_{i=k+1}^{\ell} c_i + \mathbb{I}_{\ell < |F|} \cdot c'_{\ell+1}} + 1 \\
&\underset{(11)}{\geq} \frac{\sum_{i=k+1}^{\ell} w_i + \mathbb{I}_{\ell < |F|} \cdot w'_{\ell+1}}{B} + 1 \\
&\underset{(12)}{\geq} \frac{\sum_{i=1}^{k+1} w_i}{B} + 1 = \frac{\sum_{i=1}^{k+1}(v_i - c_i)}{B} + 1 \\
&= \frac{\sum_{i=1}^{k+1} v_i}{B} + \frac{B - \sum_{i=1}^{k+1} c_i}{B} \\
&\underset{(13)}{\geq} \frac{\sum_{i=1}^{k+1} v_i}{B}.
\end{aligned}
$$

In the above, (10) follows Lemma 6, (11) and (13) follow from $\sum_{i=1}^{\ell} c_i + \mathbb{I}_{\ell < |F|} \cdot c'_{\ell+1} \leq B$, and (12) follows Eq. (9). Rearranging Eq. (13) gives

$$
c_{k+1}/v_{k+1} < \frac{B}{\sum_{i=1}^{k+1} v_i},
$$

implying Mechanism 1 would have added $k+1$ to $W$ as well, contradicting $k$'s maximality. $\qquad\square$

Using the above lemma, we're able to prove the welfare guarantees of the mechanism.

**Theorem 3.** *The welfare obtained by Mechanism 1 is a $\frac{1}{2+\sqrt{2}}$-approximation to the optimal welfare.*

*Proof.* Recall that

$$
\text{FOpt}' = \text{FOpt}(\{(w_i, c_i)\}_{i \in F \setminus \{i^*\}}, B).
$$

Let ALG denote the welfare obtained by Mechanism 1 and OPT denote the optimal welfare. Moreover, with slight abuse of notation, let $\text{FOpt} = \text{FOpt}(\{(w_i, c_i)\}_{i \in F}, B)$ be the fractional optimal solution with all agents (agents not in $F$ can only decrease the objective).

First, consider the case where $i^*$ was allocated, which implies that $w_{i^*} \geq \text{FOpt}'/(1+\sqrt{2})$. We have that

$$
\begin{aligned}
(2+\sqrt{2})\text{ALG} &= (1+\sqrt{2})w_{i^*} + w_{i^*} \\
&\geq \text{FOpt}' + w_{i^*} \geq \text{FOpt} \\
&\geq \text{OPT}.
\end{aligned}
$$

For the case $W$ was allocated, recall that

$$
\text{FOpt} = \sum_{i \leq \ell} w_i + \mathbb{I}_{\ell < |F|} \cdot w'_{\ell+1}.
$$

If $\ell > k$, we bound OPT by

$$
\begin{aligned}
\text{OPT} &\leq \text{FOpt} = \sum_{i \leq \ell} w_i + \mathbb{I}_{\ell < |F|} \cdot w'_{\ell+1} \\
&= \sum_{i \leq k} w_i + \sum_{k+1 \leq i \leq \ell} w_i + \mathbb{I}_{\ell < |F|} \cdot w'_{\ell+1} \\
&\leq 2 \cdot \sum_{i \leq k} w_i + w_{k+1} \leq 2 \cdot \text{ALG} + w_{i^*} \\
&\leq 2 \cdot \text{ALG} + \text{OPT}/(1+\sqrt{2}),
\end{aligned}
$$

where the second inequality follows Lemma 7. Rearranging gives the desired bound.

When $k = \ell$, we get even a stronger bound since

$$
\begin{aligned}
\text{OPT} \quad \leq \quad \text{FOPT} &= \sum_{i \leq \ell} w_i + \mathbb{I}_{\ell < |F|} \cdot w'_{\ell+1} \\
&= \sum_{i \leq k} w_i + \mathbb{I}_{\ell < |F|} \cdot w'_{\ell+1} \\
&\leq \quad \text{ALG} + w_{i^*} \leq \text{ALG} + \text{OPT}/(1 + \sqrt{2}).
\end{aligned}
$$

Thus, we show the desired bound. $\qquad \square$

### B.3. Generalized Welfare Maximization

We consider the problem of maximizing a generalized objective subject to a hard budget constraint $B$. Let each agent $i$ have a public valuation $v_i$ and a private cost $c_i$. We define the weight of agent $i$ as $w_i = v_i - \alpha c_i$ for a known scalar $\alpha \geq 0$. Our goal is to maximize $G(\sum_{i \in S} w_i)$, where $G : \mathbb{R}_{\geq 0} \to \mathbb{R}_{\geq 0}$ is a non-decreasing, concave and normalized ($G(0) = 0$) function.

Our mechanism is an adaptation of the welfare-maximizing mechanism that incorporates the parameter $\alpha$ into the selection criteria while maintaining budget feasibility via value-based thresholding.

As before, $\text{FOPT}(\cdot, \cdot)$ denote the value of the optimal fractional solution to the Knapsack problem given a set of weight-cost tuples and capacity $B$.

---

**Mechanism 3** Generalized-Welfare Budget Feasible Auction

---

1: **Input:** Reported costs $\mathbf{c} = (c_1, \ldots, c_n)$, Known valuations $\mathbf{v} = (v_1, \ldots, v_n)$, Budget $B$, parameter $\alpha > 0$
2: **Output:** Allocation set and payments

3: Let $F = \{i : c_i \leq \min(v_i/\alpha, B)\}$        ▷ Restrict to sellers with positive welfare and feasible cost.
4: Rename bidders in $F$ such that $\frac{v_1}{c_1} \geq \frac{v_2}{c_2} \geq \ldots \geq \frac{v_{|F|}}{c_{|F|}}$.
5: Set $W \leftarrow \emptyset$, $k \leftarrow 1$

6: **while** $k \leq |F|$ and $\frac{c_k}{v_k} \leq \frac{B}{\sum_{j \leq k} v_j}$ **do**
7:  $W \leftarrow W \cup \{k\}$
8:  $k \leftarrow k + 1$
9: **end while**

10: For every $i \in F$, let $w_i = v_i - \alpha \cdot c_i$

11: Let $i^* \in \arg\max_{i \in F} w_i$
12: Compute $\text{FOPT}' = \text{FOPT}(\{(w_i, c_i)\}_{i \in F \setminus \{i^*\}}, B)$   ▷ Compute the optimal fractional solution of all agents but $i^*$.
13: **if** $w_{i^*} > \text{FOPT}'/(1 + \sqrt{2})$ **then**
14:  Allocate $\{i^*\}$                   ▷ Allocate $i^*$ if $i^*$ has high welfare
15: **else**
16:  Allocate the set $W$                 ▷ Otherwise, allocate $W$
17: **end if**

18: Pay agents using Myersonian payments

---

### B.4. Analysis

As the analysis of the generalized mechanism closely resembles the analysis for the welfare objective, we shortly state how the proof would adapt to this case.

**Theorem 13** (Budget Feasibility and Truthfulness). *Mechanism 3 is DSIC, IR, and Ex-Post Budget Feasible.*

*Proof.* The proof of monotonicity is identical to that in Theorem 2, which implies DSIC-IR as we use Myersonian payments. An identical proof to that of Lemma 5 establishes that when $W$ is allocated, every agent $i \in W$ receives payment of at most $T_i = v_i \cdot \frac{B}{\sum_{j \in W} v_j}$, implying a total payment of at most $\sum_{i \in W} T_i \leq B$. When $i^*$ is the only agent allocated, then as we filter agents with cost larger than $B$, budget-feasibility is immediate. $\square$

**Theorem 14** (Approximation Guarantee). *Mechanism $\mathcal{M}^\alpha$ provides a $\frac{1}{2+\sqrt{2}}$-approximation for the objective $G(\sum_{i \in S} w_i)$.*

*Proof.* We first consider the objective of maximizing $\sum_i w_i$ subject to budget-feasibility, where OPT is the optimal sum of weights, and ALG is the sum of weights of agents allocated by our mechanism. We analogously define $k$ and $\ell$ as in Appendix B.2. The proof of Lemma 6 is identical, as the sorting of weights-over-costs is identical to the sorting of values-over-costs. The statement of Lemma 7 stays the same, but the proof needs minor adaptations, as now $v_{k+1}/c_{k+1} = w_{k+1}/c_{k+1} + \alpha$, and applying the contradictory assumption and budget feasibility yields that

$$
\begin{aligned}
v_{k+1}/c_{k+1} \quad &> \quad \frac{\sum_{i=1}^{k+1} v_i}{B} + \frac{\alpha(B - \sum_{i=1}^{k+1} c_i)}{B} \\
&\geq \quad \frac{\sum_{i=1}^{k+1} v_i}{B}.
\end{aligned}
$$

As in Lemma 7, rearranging shows that agent $k+1$ should have been added to $W$, implying a contradiction.

With Lemmas 6 and 7, the proof of approximation holds, and we can show $\text{ALG} \geq \text{OPT}/(2 + \sqrt{2})$. In order to generalize to concave, monotone and normalized objective $G(\sum_i w_i)$, we note that since $G$ is monotone, $G(\text{OPT})$ is the optimal solution for the new objective. The value our mechanism obtains is:

$$
\begin{aligned}
G(\text{ALG}) &\geq G(\text{OPT}/(2+\sqrt{2})) \\
&\geq G(\text{OPT})/(2+\sqrt{2}),
\end{aligned}
$$

where the last inequality holds because $G$ is concave and $G(0) = 0$. $\square$

### B.5. Prior-free Inapproximability of Buyer's Utility

**Proposition 6.** *In the prior-free setting, optimal utility is inapproximable even for a single agent.*

*Proof.* By Lemma 1, any DSIC-IR mechanism must be monotone. For a single agent, a monotone allocation rule corresponds to setting a deterministic price threshold $p$. The mechanism offers to buy the item at price $p$. If the seller reports a cost $c \leq p$, the transaction occurs at price $p$; otherwise, it does not.

To satisfy the budget constraint and to ensure non-negative utility, the threshold must satisfy $p \leq B = 1$. The buyer's utility is defined as $v - p = 1 - p$ if the trade occurs, and $0$ otherwise.

We demonstrate that for any choice of threshold $p$, there exists a valid cost $c$ such that the mechanism achieves zero utility while the optimal benchmark achieves positive utility:

1. **Case 1: The mechanism sets a threshold $p < 1$.** Suppose the seller's true cost is $c = \frac{1+p}{2}$. Notice that $p < c < 1$. Since the seller's cost $c$ is strictly greater than the offer $p$, the seller rejects the trade, and the mechanism's utility is $0$. However, trade is efficient because $c < v$. An optimal benchmark (with knowledge of $c$) could offer a price equal to $c$, resulting in a strictly positive utility of $1 - c > 0$.

2. **Case 2: The mechanism sets a threshold $p = 1$.** Suppose the seller's true cost is $c = 0$. Since $c \leq p$, the seller accepts the trade. The buyer pays $p = 1$ for an item of value $v = 1$, resulting in a net utility of $1 - 1 = 0$. However, an optimal benchmark could offer a price $p' = \epsilon$ (where $\epsilon > 0$ is arbitrarily small), securing the item for nearly zero cost and achieving a utility of $1 - \epsilon \approx 1$.

In both cases, the ratio of the optimal utility to the mechanism's utility is unbounded. Thus, no truthful mechanism can approximate the optimal utility. $\square$

## C. Additional Material from Section 4

### C.1. Missing Proofs from Subsection 4.1

We will rewrite program Equation (3) using standard Bayesian mechanism-design tools. Myerson (1981) dictates that to satisfy DSIC and IR, the mechanism needs to have an allocation function $\mathbf{x}(\cdot)$ that is monotone *non-increasing* in cost, and a payment function $\mathbf{p}(\cdot)$ that satisfies the Myersonian formula (from Lemma 1). Furthermore, we can rewrite the expected payment to an agent $i$ using the virtual cost functions $\phi_i(c_i) = c_i + \frac{F_i(c_i)}{f_i(c_i)}$ and the allocation function $\mathbf{x}(\cdot)$ as

$$\mathbb{E}_{\mathbf{c}}[p_i(\mathbf{c})] = \mathbb{E}_{\mathbf{c}}[\phi_i(c_i) \cdot x_i(\mathbf{c})].$$

Using these tools, we reformulate the optimization problem in terms of only one variable, the allocation rule $x(\cdot)$:

$$
\begin{aligned}
\max_{x} \ & \mathbb{E}_{\mathbf{c}}\left[\sum_{i\in[n]}(v_i - \phi_i(c_i))\cdot x_i(\mathbf{c})\right] \\
\text{s.t. } & \mathbb{E}_{\mathbf{c}}\left[\sum_{i\in[n]}\phi_i(c_i)x_i(\mathbf{c})\right] \leq B, \\
& x_i(c_i, \mathbf{c}_{-i}) \geq x_i(c_i', \mathbf{c}_{-i}) \qquad \forall c_i \leq c_i', \forall \mathbf{c}_{-i}, \forall i \in [n] \\
& 0 \leq x_i(\mathbf{c}) \leq 1 \qquad\qquad\qquad \forall \mathbf{c}, \forall i \in [n].
\end{aligned}
\tag{14}
$$

**Theorem 7.** *Assuming regular cost distributions, there exists a utility-optimal mechanism $\mathcal{M}_{ante}$ for the soft-budget constraint that independently posts prices $\hat{c}_i$ to each agent $i$. Furthermore, these optimal prices $\hat{c}_i$ can be computed efficiently.*

*Proof.* The proof parallels the one in (Balkanski & Hartline, 2016), which is closely related to the one in (Ensthaler & Giebe, 2014), and is presented for completeness.

The optimal (randomized) allocation rule $x(\cdot)$ induces a distribution over sets of allocated agents $S$; call this distribution $\mathcal{D}$. Then let $\mathbf{q}$ be the marginal probability that each agent is allocated, i.e., $q_i = \Pr_{S\sim\mathcal{D}}[i \in S]$. We now state a crucial lemma from (Balkanski & Hartline, 2016) that restricts the optimal mechanism.

**Lemma 8.** *Let agent $i$ have cost $c_i$ drawn from a regular distribution $F_i$. For any incentive-compatible mechanism that allocates to agent $i$ with (ex-ante) probability $q_i$, the expected payment paid to agent $i$ is at least $q_i\hat{c}_i$, where $\hat{c}_i = F_i^{-1}(q_i)$.*

Unlike the value objective, under utility maximization we must address whether the optimal solution necessarily expends the entire budget. Exploiting the structure of (14) together with the regularity of the virtual cost functions, we can solve $v_i = \phi_i(c_i)$ to identify, for each agent, the price beyond which offering a higher price would decrease the objective. Let $\bar{c}_i$ denote this price and let $\bar{q}_i := F_i(\bar{c}_i)$ be the probability that agent $i$ accepts it. This yields an upper bound on total useful expenditure, $\bar{B} := \sum_i \bar{q}_i \bar{c}_i$. If the available budget satisfies $B \leq \bar{B}$, then the optimal mechanism expends the entire budget; if $B > \bar{B}$, we can equivalently solve the problem assuming that the budget is $\bar{B}$. Consequently, without loss of generality, we may restrict attention to instances in which the budget binds for utility maximization as well.

Lemma 8 implies that among all incentive-compatible mechanisms that induce the same marginal allocation probabilities $\mathbf{q}$, independent posted pricing minimizes the expected payments. Consequently, for any feasible mechanism, there exists an independent posted-price mechanism that induces the same marginals, expends weakly less budget, and achieves the same expected utility. As a result, the mechanism design problem reduces to optimizing directly over marginal allocation probabilities $\mathbf{q}$, with the understanding that any optimal choice of marginals can be implemented by an independent posted-price mechanism.

Following the methodology of (Balkanski & Hartline, 2016), we adopt a Lagrangian virtual-surplus framework for Bayesian budget-feasible mechanism design. In contrast to their value objective, our objective is expected *utility*. Introducing a Lagrange multiplier $\lambda \geq 0$ for the budget constraint, the resulting Lagrangian objective is

$$\max_{x} \; \lambda B + \sum_i \mathbb{E}_{\mathbf{c}}\big[\big(v_i - (\lambda + 1)\phi_i(c_i)\big) \, x_i(\mathbf{c})\big].$$

For any fixed $\lambda$, this objective decomposes pointwise and is maximized by selecting agent $i$ whenever $v_i \geq (\lambda + 1)\phi_i(c_i)$, which assuming regularity induces an independent monotone allocation rule with threshold $\hat{c}_i = \phi_i^{-1}\big(v_i/(\lambda+1)\big)$, that is, a posted price. As such, the maximizer of the Lagrangian objective satisfies IC and IR as well, so it is also a maximizer of the original program.

Let $\hat{q}_i = F_i(\hat{c}_i)$ denote the acceptance probability of the price $\hat{c}_i$. The expected payment is $\sum_i \hat{c}_i \hat{q}_i$, which is monotonically decreasing in $\lambda$. Hence, we can (efficiently) binary search for the Lagrange multiplier for which $\sum_i \hat{c}_i \hat{q}_i = B$.

$\square$

### C.2. Missing Proofs from Subsection 4.2

We now present the proof of Lemma 3.

**Lemma 3.** *Assuming prices $\mathbf{p}$ that satisfy the budget constraint ex-ante, are individually ex-post feasible ($p_i \leq B$) and additionally satisfy $p_i \geq \frac{B}{\alpha}$, the sequential posted pricing mechanism that orders agents by decreasing $v_i - p_i$ and offers them prices $p_i$, achieves a $\frac{1-e^{-\alpha}}{\alpha}$-approximation to the ex-ante posted pricing utility $U_{price}(\mathbf{p})$.*

*Proof.* We will lower bound the expected utility $U$ of this mechanism by the expected value of the first agent that accepts their offered price. Without loss of generality, assume that agents are ordered in decreasing expected utility $u_i = v_i - p_i$. Define $\Delta_\ell := u_\ell - u_{\ell+1} \geq 0$ with $u_{n+1} = 0$, and let $Q = \sum_i^n q_i$ and $Q^\ell = \sum_{i=1}^\ell q_i$. We rewrite the ex-ante posted pricing utility as:

$$U_{\text{price}}(\mathbf{p}) = \sum_{i=1}^n u_i q_i = \sum_{i=1}^n \left(\sum_{\ell \geq i} \Delta_\ell\right) \cdot q_i = \sum_{\ell=1}^n \Delta_\ell \cdot \sum_{i \leq \ell} q_i = \sum_{\ell=1}^n \Delta_\ell \cdot Q^\ell.$$

Since, by assumption, these prices satisfy ex-ante budget feasibility:

$$B \geq \sum_{i=1}^n q_i p_i \geq \frac{B}{\alpha} \sum_{i=1}^n q_i = \frac{B}{\alpha} Q \quad \Rightarrow \quad Q \leq \alpha,$$

where the second inequality is by the Lemma's assumption.

We now lower bound the utility of the mechanism with the expected utility of the first agent to accept their price:

$$
\begin{aligned}
U(\mathbf{p}) &\geq \sum_{i=1}^n u_i q_i \prod_{j < i}(1 - q_j) \\
&= \sum_{\ell=1}^n \Delta_\ell \sum_{i \leq \ell} q_i \prod_{j < i}(1 - q_j) && \text{(rewrite using } \Delta_\ell) \\
&= \sum_{\ell=1}^n \Delta_\ell \left(1 - \prod_{j \leq \ell}(1 - q_j)\right) && \text{(rewrite probability using the complement event)} \\
&\geq \sum_{\ell=1}^n \Delta_\ell \left(1 - e^{-\sum_{j \leq \ell} q_j}\right) && 1 - x \leq e^{-x} \\
&= \sum_{\ell=1}^n \Delta_\ell \left(1 - e^{-Q^\ell}\right) && \text{(definition of } Q^\ell).
\end{aligned}
$$

Combining all three inequalities, we get:

$$
\begin{aligned}
U(\boldsymbol{p}) &\geq \frac{\sum_\ell \Delta_\ell (1 - e^{-Q^\ell})}{\sum_\ell \Delta_\ell \cdot Q^\ell} \cdot U_{\text{price}}(\boldsymbol{p}) \\
&\geq \min_\ell \frac{(1 - e^{-Q^\ell})}{Q^\ell} \cdot U_{\text{price}}(\boldsymbol{p}) \\
&\geq \frac{1 - e^{-Q}}{Q} \cdot U_{\text{price}}(\boldsymbol{p}) \\
&\geq \frac{1 - e^{-\alpha}}{\alpha} \cdot U_{\text{price}}(\boldsymbol{p}),
\end{aligned}
\tag{15}
$$

where the fourth inequality follows because $\frac{1-e^{-x}}{x}$ is a decreasing function in $x$.

$\square$

We now present the proof of the approximation guarantee of the standalone mechanism $\mathcal{M}$.

**Lemma 2.** *Assuming prices $\boldsymbol{p}$ that satisfy the budget constraint ex-ante and are individually ex-post feasible ($p_i \leq B$), $\mathcal{M}(\boldsymbol{p})$ achieves a $(1 - \frac{1}{\sqrt{2\pi k}})(1 - \frac{1}{k})$-approximation to the ex-ante posted pricing utility $U_{price}(\boldsymbol{p})$.*

*Proof.* For the analysis, let $N$ denote the agents ordered in decreasing ratio $(v_i - p_i)/p_i$.

In this proof of Lemma 2, we will use the concept of "*fractional*-knapsack utility". If we fix the payment to each agent $i$ as $p_i$, then our utility maximization problem is just a knapsack optimization problem, constrained by our budget $B$ and payments. However, if each payment is accepted with (independent) ex-ante probability $q_i$, and we allow for fractional allocations, this is instead the independent fractional knapsack problem, which admits a greedy solution: serve agents in decreasing order of utility-per-payment $\frac{v_i - p_i}{p_i}$. The utility of this fractional solution is what we call *fractional*-knapsack utility, and is formally defined as:

$$
\begin{aligned}
U_B(\boldsymbol{p}) &= \mathbb{E}_{S \sim ind(\boldsymbol{q})}\left[u_B(S)\right] \\
&= \mathbb{E}_{S \sim ind(\boldsymbol{q})}\left[\sum_{i \in N} \frac{v_i - p_i}{p_i}\left(\min\left\{B, \sum_{j \in S \cap \{1,\ldots,i\}} p_j\right\} - \min\left\{B, \sum_{j \in S \cap \{1,\ldots,i-1\}} p_j\right\}\right)\right],
\end{aligned}
\tag{16}
$$

where $S$ denotes the *realized* set of agents that will accept their price should they be offered it and $ind(\boldsymbol{q})$ the product distribution, where each agent is realized independently with probability $q_i$.

Let us denote the expected utility of the mechanism $\mathcal{M}$ as $U_{\mathcal{M}}(\boldsymbol{p}) = \mathbb{E}_{S \sim ind(\boldsymbol{q})}[u_{\mathcal{M}}(S)]$. This expected utility is almost equal to the optimal independent *fractional*-knapsack utility $U_B(\boldsymbol{p})$ (losing out only on the contribution of the (at most) one agent that would be fractionally served). We first prove this pointwise, for a fixed set $S$. Let the fractional agent be agent $\ell$, and them being served fractionally at $\gamma \in [0, 1)$. We lower bound the *fractional*-knapsack utility $u_B(S)$:

$$
u_B(S) = \sum_{i \in S \cap \{1,\ldots,\ell-1\}} \frac{v_i - p_i}{p_i} \cdot p_i + \gamma \cdot \frac{v_\ell - p_\ell}{p_\ell} \cdot p_\ell \geq \frac{v_\ell - p_\ell}{p_\ell}\left(\sum_{i \in S \cap \{1,\ldots,\ell-1\}} p_i + \gamma p_\ell\right) = \frac{v_\ell - p_\ell}{p_\ell} \cdot B, \tag{17}
$$

where the inequality uses the fact that agents are ordered by decreasing $\frac{v_i - p_i}{p_i}$, and the last equality uses the fact that for a fractional agent to exist, the expenditure for all agents should be equal to the budget $B$.

We now lower bound the mechanism's utility using the *fractional*-knapsack utility:

$$
u_{\mathcal{M}}(S) \geq u_B(S) - \frac{v_\ell - p_\ell}{p_\ell} \cdot p_\ell \geq \left(1 - \frac{p_\ell}{B}\right) \cdot u_B(S) \geq \left(1 - \frac{1}{k}\right) \cdot u_B(S), \tag{18}
$$

where the first inequality is because $\mathcal{M}$ loses out at most the utility of the fractional agent compared to the *fractional*-knapsack utility, the second inequality is due to Equation (17), and the third inequality is due to the $p_\ell \leq \frac{B}{k}$.

Taking expectations over the set $S$ yields the desired lower bound for mechanism's $\mathcal{M}$ expected utility:

$$U_\mathcal{M}(\boldsymbol{p}) \geq \left(1 - \frac{1}{k}\right) U_B(\boldsymbol{p}) \tag{19}$$

We now focus on the performance of the optimal independent *fractional*-knapsack utility $U_B(\boldsymbol{p})$. We can rewrite $U_B(\boldsymbol{p})$ starting from its definition (16) as:

$$
\begin{aligned}
U_B(\boldsymbol{p}) &= \mathbb{E}_{S \sim ind(\boldsymbol{q})}[u_B(S)] \\
&= \mathbb{E}_{S \sim ind(\boldsymbol{q})}\left[\sum_{i \in N} \frac{v_i - p_i}{p_i} \left(\min\left\{B, \sum_{j \in S \cap \{1,\ldots,i\}} p_j\right\} - \min\left\{B, \sum_{j \in S \cap \{1,\ldots,i-1\}} p_j\right\}\right)\right] \\
&= \sum_{i \in N}\left(\left(\frac{v_i - p_i}{p_i} - \frac{v_{i+1} - p_{i+1}}{p_{i+1}}\right) \cdot \mathbb{E}_{S \sim ind(\boldsymbol{q})}\left[\min\left(B, \sum_{j \in S \cap \{1,\ldots,i\}} p_j\right)\right]\right).
\end{aligned}
\tag{20}
$$

The third equality follows from rearranging the terms in a telescopic sum, using linearity of expectation, and using the convention that $\frac{v_{n+1} - p_{n+1}}{p_{n+1}} = 0$.

Now, for our benchmark, we will be comparing the performance of our mechanism to $U_{\mathrm{price}}(\boldsymbol{p})$, the ex-ante posted pricing utility of the prices $\boldsymbol{p}$. The ex-ante posted pricing utility can be rewritten as:

$$
\begin{aligned}
U_{\mathrm{price}}(\boldsymbol{p}) = \mathbb{E}_{S \sim ind(\boldsymbol{q})}[u(S)] &= \mathbb{E}_{S \sim ind(\boldsymbol{q})}\left[\sum_{i \in S}(v_i - p_i)\right] \\
&= \mathbb{E}_{S \sim ind(\boldsymbol{q})}\left[\sum_{i \in S} p_i \cdot \frac{v_i - p_i}{p_i}\right] \\
&= \mathbb{E}_{S \sim ind(\boldsymbol{q})}\left[\sum_{i \in N}\left(\frac{v_i - p_i}{p_i}\right) \cdot \left(\sum_{j \in S \cap \{i\}} p_j\right)\right] \\
&= \sum_{i \in N}\left(\frac{v_i - p_i}{p_i} - \frac{v_{i+1} - p_{i+1}}{p_{i+1}}\right) \cdot \mathbb{E}_{S \sim ind(\boldsymbol{q})}\left[\sum_{j \in S \cap \{1,\ldots,i\}} p_j\right],
\end{aligned}
\tag{21}
$$

where the last equality is rearranging terms in a telescopic sum and uses linearity of expectation.

We now present a technical lemma, that we will use to lower bound the optimal independent *fractional*-knapsack utility with the ex-ante posted pricing utility (the proof of which we will present later).

**Lemma 9.** *For any budget $B > 0$, $k \geq 1$, prices $p_j \leq B/k$ and marginal probabilities $q_j$ that satisfy the ex-ante budget constraint $\sum q_j \cdot p_j \leq B$, we prove*

$$
\frac{\mathbb{E}_{S \sim ind(\boldsymbol{q})}\left[\min\left(B, \sum_{j \in S \cap \{1,\ldots,i\}} p_j\right)\right]}{\mathbb{E}_{S \sim ind(\boldsymbol{q})}\left[\sum_{j \in S \cap \{1,\ldots,i\}} p_j\right]} \geq \frac{\mathbb{E}_{S \sim ind(\boldsymbol{q}')}\left[\min\left(B, \sum_{j \in S \cap \{1,\ldots,i\}} \frac{B}{k}\right)\right]}{\mathbb{E}_{S \sim ind(\boldsymbol{q}')}\left[\sum_{j \in S \cap \{1,\ldots,i\}} \frac{B}{k}\right]} \geq \left(1 - \frac{1}{\sqrt{2\pi k}}\right),
$$

*where $q_j' = \frac{k}{B} \cdot p_j \cdot q_j$.*

Using this Lemma 9, we now compare the optimal independent *fractional*-knapsack utility $U_B(\boldsymbol{p})$ to the benchmark $U_{\mathrm{price}}(\boldsymbol{p})$:

$$
\begin{aligned}
U_B(\boldsymbol{p}) &= \sum_{i \in N} \left( \left( \frac{v_i - p_i}{p_i} - \frac{v_{i+1} - p_{i+1}}{p_{i+1}} \right) \cdot \mathbb{E}_{S \sim ind(\boldsymbol{q})} \left[ \min \left( B, \sum_{j \in S \cap \{1, \ldots, i\}} p_j \right) \right] \right) \\
&\geq \sum_{i \in N} \left( \left( \frac{v_i - p_i}{p_i} - \frac{v_{i+1} - p_{i+1}}{p_{i+1}} \right) \cdot \left( 1 - \frac{1}{\sqrt{2\pi k}} \right) \cdot \mathbb{E}_{S \sim ind(\boldsymbol{q})} \left[ \sum_{j \in S \cap \{1, \ldots, i\}} p_j \right] \right) \\
&= \left( 1 - \frac{1}{\sqrt{2\pi k}} \right) \cdot U_{\text{price}}(\boldsymbol{p})
\end{aligned}
\tag{22}
$$

We now express mechanism's $\mathcal{M}$ expected utility guarantee:

$$
U_{\mathcal{M}}(\boldsymbol{p}) \geq \left( 1 - \frac{1}{k} \right) \cdot U_B(\boldsymbol{p}) \geq \left( 1 - \frac{1}{k} \right) \cdot \left( 1 - \frac{1}{\sqrt{2\pi k}} \right) \cdot U_{\text{price}}(\boldsymbol{p}),
$$

where the first inequality is due to Equation (19), the second is due to Equation (22).

$\square$

We now present the proof of the technical Lemma 9.

*Proof.* Define:

$$
L_i = \frac{\mathbb{E}_{S \sim ind(\boldsymbol{q})} \left[ \min \left( B, \sum_{j \in S \cap \{1, \ldots, i\}} p_j \right) \right]}{\mathbb{E}_{S \sim ind(\boldsymbol{q})} \left[ \sum_{j \in S \cap \{1, \ldots, i\}} p_j \right]}, \qquad L_i' = \frac{\mathbb{E}_{S \sim ind(\boldsymbol{q}')} \left[ \min \left( B, \sum_{j \in S \cap \{1, \ldots, i\}} \frac{B}{k} \right) \right]}{\mathbb{E}_{S \sim ind(\boldsymbol{q}')} \left[ \sum_{j \in S \cap \{1, \ldots, i\}} \frac{B}{k} \right]}.
$$

We first prove that $L_i \geq L_i'$, and then prove that $L_i' \geq \left( 1 - \frac{1}{\sqrt{2\pi k}} \right)$.

**Proving $L_i \geq L_i'$.** We prove the first inequality by an iterative coordinate replacement argument. Fix an index $\ell \leq i$ and replace $(q_\ell, p_\ell)$ by $\left( q_\ell', \frac{B}{k} \right)$, where $q_\ell' := \frac{k}{B} \cdot p_\ell \cdot q_\ell$, while keeping all other coordinates fixed. Note that by this transformation $q_\ell' \leq q_\ell \leq 1$.

Notice that this replacement preserves the *denominator*, since $q_\ell' \cdot \frac{B}{k} = q_\ell p_\ell$, and all other terms remain unchanged.

For the *numerator*, define function $h(x) = \min(B, t + x) - \min(B, t)$, which for any fixed constant $t \geq 0$ is concave in $x$. Now, condition on the realization of all other marginals besides the $\ell$-th one, which fixes set $S$ (besides the $\ell$-th agent). This fix induces some $t$ such that $\sum_{j \in S \cap \{1, \ldots, i\} \setminus \{\ell\}} p_j = t$. We now quantify the expected contribution of agent $\ell$ with parameters $(q_\ell, p_\ell)$ and with parameters $\left( q_\ell', \frac{B}{k} \right)$:

$$
\begin{aligned}
\mathbb{E}_{q_\ell} \left[ \min(B, t + p_\ell) \,\Bigg|\, \sum_{j \in S \cap \{1, \ldots, i\} \setminus \{\ell\}} p_j = t \right] &= q_\ell \cdot \min(B, t + p_\ell) + (1 - q_\ell) \cdot \min(B, t) \\
&= q_\ell h(p_\ell) + \min(B, t) \\
&\geq q_\ell' h\left( \frac{B}{k} \right) + \min(B, t) \\
&= \mathbb{E}_{q_\ell'} \left[ \min\left( B, t + \frac{B}{k} \right) \,\Bigg|\, \sum_{j \in S \cap \{1, \ldots, i\} \setminus \{\ell\}} p_j = t \right],
\end{aligned}
$$

where we have used the concavity property $h(\lambda x + (1 - \lambda) y) \geq \lambda h(x) + (1 - \lambda) h(y)$ with $x = \frac{B}{k}$, $y = 0$, $\lambda = \frac{k}{B} \cdot p_\ell$ and $h(0) = 0$.

Taking expectations over the realizations of $t$ (or equivalently $S$) proves the claim for a single coordinate change, and iteratively applying this approach for all coordinates yields the desired inequality.

As a result, we have proven the *first inequality* of the lemma $L_i \geq L_i'$, as required.

**Proving** $L_i' \geq \left(1 - \frac{1}{\sqrt{2\pi k}}\right)$. We now lower-bound $L_i'$. Since $p_j' = \frac{B}{k}$ for all $j$, we will show that $L_i'$ can be expressed as the correlation-gap ratio of a $k$-highest-value-elements set function, and then invoke Theorem 15.

**Theorem 15.** *(Yan, 2011)* *For a $k$-highest-value-elements set function $f(\cdot)$, which is additive with value $f_i$ for element $i$ up to a capacity of at most $k$ elements, the correlation gap $CG(f)$ satisfies*

$$CG(f) = \inf_{\boldsymbol{q} \in [0,1]^N} \frac{\mathbb{E}_{S \sim ind(\boldsymbol{q})}[f(S)]}{\max_{\mathcal{D} \in corr(\boldsymbol{q})} \mathbb{E}_{S \sim \mathcal{D}}[f(S)]} \geq 1 - \frac{1}{\sqrt{2\pi k}},$$

*where $corr(\boldsymbol{q})$ is the set of all distributions with marginals $\boldsymbol{q}$.*

Define the set function:

$$f_i(S) := \min\left(B, \sum_{j \in S \cap \{1,\ldots,i\}} \frac{B}{k}\right) = \frac{B}{k} \cdot \min\left(k, |S \cap \{1,\ldots,i\}|\right).$$

The numerator of $L_i'$ is $\mathbb{E}_{S \sim ind(\boldsymbol{q}')}[f_i(S)]$, so we will show that the denominator of $L_i'$ equals to $\max_{\mathcal{D} \in corr(\boldsymbol{q}')} \mathbb{E}_{S \sim \mathcal{D}}[f_i(S)]$. For any (possible correlated) distribution $\mathcal{D}$ with marginals $\boldsymbol{q}'$, we have pointwise $f_i(S) \leq \sum_{j \in S \cap \{1,\ldots,i\}} p_j'$, and therefore

$$\mathbb{E}_{S \sim \mathcal{D}}[f_i(S)] \leq \mathbb{E}_{S \sim \mathcal{D}}\left[\sum_{j \in S \cap \{1,\ldots,i\}} p_j'\right] = \sum_{j \leq i} p_j' \cdot \Pr_{S \sim \mathcal{D}}[j \in S] = \sum_{j \leq i} q_j' p_j'. \tag{23}$$

On the other hand, since $\sum_{j \in N} q_j \cdot p_j = \sum_{j \in N} q_j' \cdot \frac{B}{k} \leq B$, we have that $\sum_{j \leq i} q_j' \leq k$. The polytope $P = \{q \in \mathbb{R}^n : q_i \in [0,1], \sum_{i=1}^n q_i \leq k\}$ is the convex hull of integral (full-rank) vectors that satisfy $\sum_i q_i \leq k$. Hence, by Caratheodory's theorem, every vector in $P$ can be expressed as a convex combination of these corner integral vectors. Hence there exists a distribution $\mathcal{D}^* \in corr(\boldsymbol{q}')$ supported only on sets $S \subseteq \{1,\ldots,i\}$ with $|S| \leq k$ and having marginals $\boldsymbol{q}_{\leq i}'$. For any such $S$ the budget never binds ex post, and therefore $f_i(S) = \sum_{j \in S \cap \{1,\ldots,i\}} p_j'$ holds pointwise. Thus:

$$\mathbb{E}_{S \sim \mathcal{D}^*}[f_i(S)] = \mathbb{E}_{S \sim \mathcal{D}^*}\left[\sum_{j \in S \cap \{1,\ldots,i\}} p_j'\right] = \sum_{j \leq i} p_j' \cdot q_j' = \mathbb{E}_{S \sim ind(\boldsymbol{q}')}\left[\sum_{j \in S \cap \{1,\ldots,i\}} p_j'\right]. \tag{24}$$

Combining the equality (24) with inequality (23) shows that

$$\max_{\mathcal{D} \in corr(\boldsymbol{q}')} \mathbb{E}_{S \sim \mathcal{D}}[f_i(S)] = \mathbb{E}_{S \sim ind(\boldsymbol{q}')}\left[\sum_{j \in S \cap \{1,\ldots,i\}} p_j'\right],$$

and thus $L_i'$ coincides with the correlation-gap ratio of $f_i(\cdot)$.

Finally, since $f_i(\cdot)$ is a $k$-highest-value-elements set function, Theorem 15 implies

$$L_i' \geq 1 - \frac{1}{\sqrt{2\pi k}}.$$

$\square$

**Remark 1** (Correction to (Balkanski & Hartline, 2016) Analysis). The methodology we have employed in proving Lemmas 2 and 9 uses the same tools presented in Balkanski & Hartline (2016). The crucial difference (besides of course the difference in objective) is that in our approach, we state Lemma 9 in the form of a lower bound, which we subsequently use in

Equation (22). In contrast, Balkanski & Hartline (2016) use Lemma 9 to argue the following quantity is minimized when $p_i = \frac{B}{k}$:

$$\frac{\sum_{i \in N}\left(\left(\frac{v_i}{p_i} - \frac{v_{i+1}}{p_{i+1}}\right) \cdot \mathbb{E}_{S \sim ind(\boldsymbol{q})}\left[\min\left(B, \sum_{j \in S \cap \{1,\ldots,i\}} p_j\right)\right]\right)}{\sum_{i \in N}\left(\left(\frac{v_i}{p_i} - \frac{v_{i+1}}{p_{i+1}}\right) \cdot \mathbb{E}_{S \sim ind(\boldsymbol{q})}\left[\sum_{j \in S \cap \{1,\ldots,i\}} p_j\right]\right)}.$$

In order to deduce that the global ratio is minimized at $p_i = \frac{B}{k}$ for all $i$, we would require that the coefficients are fixed (independent of $p$). As such, these arguments do not suffice to prove their claim, and so to the best of our understanding, their claim is inaccurate. Fortunately, our approach fixes this concern and nevertheless recovers the same guarantee. As such, their proposed theorem remains valid despite this unresolved step in the original reasoning.

### C.3. Missing Material from Subsection 4.3

In this subsection, we discuss ironing and showcase how it can be used to produce optimal solutions for soft-budget feasibility problem. We also provide the formal definition of mechanism $\mathcal{M}_{\text{const}}^{\text{rand}}$.

**Ironing and Irregular Distributions.** In the previous subsections, we assumed regularity to keep the exposition focused on the budget-feasibility aspect of the *ex-ante* problem. When cost distributions are irregular, we handle this with the standard ironing step: we solve the closely related *ex-ante* program defined by *ironed* virtual cost functions $\bar{\phi}_i(c_i)$ (which satisfy regularity and differ to $\phi_i(c_i)$ only on *ironed intervals*), and then translate its solution back to the original instance with minor (and standard) modifications. We define now the ironed optimization program as well as the standard lemma that ties functions $\phi_i(\cdot)$ and $\bar{\phi}_i(\cdot)$.

$$\max_x \mathbb{E}_{\mathbf{c}}\left[\sum_{i \in [n]}(v_i - \bar{\phi}_i(c_i)) \cdot x_i(\mathbf{c})\right]$$

$$\text{s.t.} \quad \mathbb{E}_{\mathbf{c}}\left[\sum_{i \in [n]}\bar{\phi}_i(c_i)x_i(\mathbf{c})\right] \leq B, \tag{25}$$

$$x_i(c_i, c_{-i}) \geq x_i(c_i', c_{-i}) \qquad \forall c_i \leq c_i', \forall c_{-i}, \forall i \in [n]$$

$$0 \leq x_i(\mathbf{c}) \leq 1 \qquad \forall \mathbf{c}, \forall i \in [n].$$

For the remainder of the subsection, as is customary in Bayesian mechanism design, we translate our problem from cost space (where seller costs are our Bayesian random variables) to quantile space, where $q = F_i(c_i)$, $c_i = F_i^{-1}(q)$. Note that higher costs correspond to higher quantiles, and thus, monotonicity of the allocation rule in cost translates to monotonicity in quantiles. We additionally express the virtual cost function as $\phi_i(q) = c_i(q) + \frac{F_i(c_i(q))}{f_i(c_i(q))} = F_i^{-1}(q) + \frac{q}{f_i(F_i^{-1}(q))}$. To facilitate the ironing procedure, we define the cumulative cost function $\Phi_i(q) = \int_0^q \phi_i(t)\, dt$ and the ironed cumulative cost function $\bar{\Phi}_i(q)$ to be the *lower* convex envelope of $\Phi_i(q)$ *pinned at the endpoints* (that is $\bar{\Phi}_i(0) = \Phi_i(0)$ and $\bar{\Phi}_i(1) = \Phi_i(1)$) and finally the ironed virtual cost function as $\bar{\phi}_i(q) = \frac{d}{dq}\bar{\Phi}_i(q)$.

We now prove the following lemma that ties the expenditure of an allocation rule, under functions $\phi_i(\cdot)$ and $\bar{\phi}_i(\cdot)$.

**Lemma 10.** *For any non-increasing allocation function $x(\boldsymbol{q})$ it holds:*

$$\mathbb{E}_{\boldsymbol{q}}[\phi_i(q_i)x(\boldsymbol{q})] \geq \mathbb{E}_{\boldsymbol{q}}[\bar{\phi}_i(q_i)x(\boldsymbol{q})], \tag{26}$$

*with equality holding as long as $x'(\boldsymbol{q}) = 0$ on all ironed interval.*

*Proof.* We express in quantiles, write the expectations with integrals and integrate by parts the difference of the functions:

$$\int_0^1 (\phi_i(q) - \bar{\phi}_i(q)) \cdot x(q)\, dq = \left[(\Phi_i(q) - \bar{\Phi}_i(q))x(q)\right]\Big|_0^1 + \int_0^1 (\Phi_i(q) - \bar{\Phi}_i(q)) \cdot (-x'(q))\, dq \geq 0.$$

Note that by construction $\Phi_i(0) = \bar{\Phi}_i(0)$ and $\Phi_i(1) = \bar{\Phi}_i(1)$, which implies that the first term vanishes. Notice that the second term is positive since $x(q)$ is non-increasing and $\Phi_i(q) - \bar{\Phi}_i(q) \geq 0$ by definition. Finally, notice that if $x'(q) = 0$ on every ironed interval, the second term is in fact 0. $\qquad \square$

We now proceed with proving the theorem about the optimal mechanism that satisfies the budget constraint ex-ante:

**Theorem 10.** *There exists a utility-optimal mechanism $\mathcal{M}_{ante}$ for the soft-budget problem that posts prices independently across agents. For each agent $i$, $\mathcal{M}_{ante}$ either (i) posts a single price $\hat{c}_i$, or (ii) randomizes between two prices by posting $\hat{c}_{i,1}$ with probability $\theta_i \in [0, 1]$ and $\hat{c}_{i,2}$ with probability $1 - \theta_i$. Moreover, the corresponding optimal prices and mixing probabilities $(\hat{\mathbf{c}}, \boldsymbol{\theta})$ can be computed efficiently.*

*Proof.* We start by comparing the optimization formulations induced by the virtual cost functions $\phi_i(\cdot)$ and the ironed virtual cost functions $\bar{\phi}_i(\cdot)$. Lemma 10 allows us to compare feasibility between (14) and (25): for any monotone allocation rule, evaluating expected expenditure with $\bar{\phi}_i$ is never more restrictive than evaluating it with $\phi_i$. Consequently, every solution feasible for the original program (14) remains feasible for the ironed one (25). In addition, for the same reason, the ironed objective weakly *upper bounds* the original objective when evaluated at the same allocation rule. Therefore, the optimal value of the ironed program upper bounds the optimal value of the original program.

Now, it suffices to show that an optimal solution of the ironed program can be modified into a feasible solution for the original program while preserving its (ironed) objective value; combining this with the above upper bound yields that this solution is optimal for the original program. Since the ironed program has *regular* virtual cost functions $\bar{\phi}_i$, its optimum decomposes across agents (as per Theorem 7) into independent posted prices $\hat{c}$; thus, it is enough to carry out this modification at the level of a *single-agent*, and then apply it independently to each agent.

Fix an agent $i$. Let $x_i^*(q)$ be the optimal solution of the ironed program (25) for agent $i$ (which corresponds to a posted price $\hat{c}_i$, or equivalently a quantile $q_i^*$). We now convert $x_i^*(\cdot)$ into a solution feasible for the original program (14) without changing its objective value. If $q_i^*$ does not fall on an ironed interval of $\bar{\phi}_i$, then Lemma 10 is already tight for $x_i^*$, and agent $i$'s expected expenditure is the same under both $\phi_i$ and $\bar{\phi}_i$.

Otherwise, $q_i^*$ falls on some ironed interval $[a, b]$ of $\bar{\phi}_i$. Note now that on an ironed interval $[a, b]$, the ironed cumulative function $\bar{\Phi}_i$ coincides with the straight-line (chord) connecting $(a, \Phi_i(a))$ and $(b, \Phi_i(b))$. Since $\bar{\Phi}_i$ is linear on $[a, b]$, its slope is constant there; hence the ironed virtual cost $\bar{\phi}_i(q)$ is constant throughout $[a, b]$.

Let $\theta_i := \frac{1}{b-a} \int_a^b x_i^*(t) \, dt \in [0, 1]$. Define $\tilde{x}_i$ by replacing the jump of $x_i^*$ inside $[a, b]$ by a constant level that preserves the total mass on that interval, as seen in Figure 1 (notice that $\tilde{x}_i$ remains non-increasing):

$$\tilde{x}_i(q) := \begin{cases} x_i^*(q), & q < a, \\ \theta_i, & a \leq q \leq b, \\ x_i^*(q), & q > b. \end{cases}$$

We rewrite agent $i$'s contribution to the (ironed) objective in quantile space:

$$\int_0^1 \left( v_i - \bar{\phi}_i(q) \right) x_i(q) \, dq = v_i \int_0^1 x_i(q) \, dq - \int_0^1 \bar{\phi}_i(q) \, x_i(q) \, dq. \tag{27}$$

We claim that $x_i^*$ evaluated under $\bar{\phi}_i$ yields the same (ironed) objective value and the same expected expenditure as $\tilde{x}_i$ evaluated under $\phi_i$. In particular, we will show the equalities Equations (32) and (33).

First, by construction $\tilde{x}_i(q) = x_i^*(q)$ for $q \notin [a, b]$. On $[a, b]$, $\tilde{x}_i$ is constant with value $\tilde{x}_i(q) = \theta_i$, and thus

$$\int_a^b \tilde{x}_i(q) \, dq = (b - a) \cdot \theta_i = (b - a) \cdot \frac{1}{b-a} \int_a^b x_i^*(t) \, dt = \int_a^b x_i^*(q) \, dq. \tag{28}$$

Next, since $\bar{\phi}_i$ is constant on the ironed interval $[a, b]$, say $\bar{\phi}_i(q) \equiv \kappa$ for $q \in [a, b]$, Equation (28) also implies

$$\int_a^b \bar{\phi}_i(q) \, \tilde{x}_i(q) \, dq = \kappa \int_a^b \tilde{x}_i(q) \, dq = \kappa \int_a^b x_i^*(q) \, dq = \int_a^b \bar{\phi}_i(q) \, x_i^*(q) \, dq. \tag{29}$$

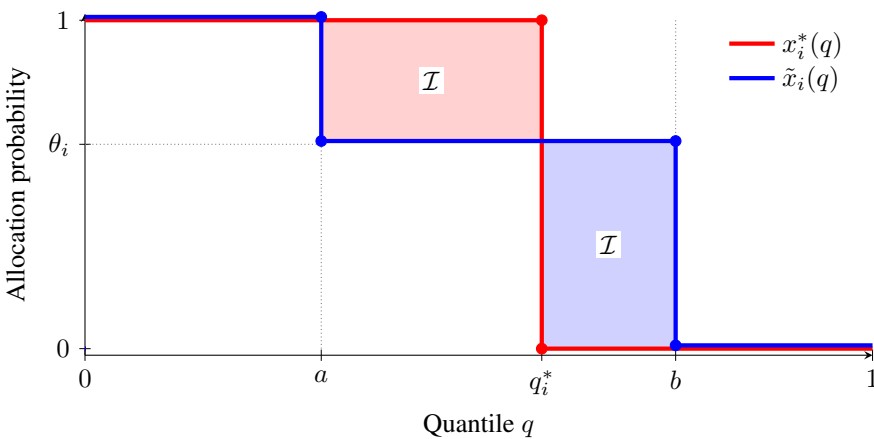

*Figure 1.* Allocation functions $\tilde{x}_i(q)$ and $x_i^*(q)$.

Since $\tilde{x}_i = x_i^*$ outside $[a, b]$, Equations (28) and (29), we get

$$\int_0^1 \tilde{x}_i(q)\,dq = \int_0^1 x_i^*(q)\,dq, \qquad \int_0^1 \bar{\phi}_i(q)\,\tilde{x}_i(q)\,dq = \int_0^1 \bar{\phi}_i(q)\,x_i^*(q)\,dq, \tag{30}$$

Plugging Equation (30) into Equation (27) (with $x_i = \tilde{x}_i$ and $x_i = x_i^*$, respectively) gives

$$\int_0^1 \left(v_i - \bar{\phi}_i(q)\right) \tilde{x}_i(q)\,dq = \int_0^1 \left(v_i - \bar{\phi}_i(q)\right) x_i^*(q)\,dq. \tag{31}$$

Next, we show that $\tilde{x}_i$ achieves the same expenditure when evaluated under $\phi_i$ or under $\bar{\phi}_i$; by tightness of Lemma 10 for $\tilde{x}_i$ and the second equality in Equation (30),

$$\int_0^1 \phi_i(q)\,\tilde{x}_i(q)\,dq = \int_0^1 \bar{\phi}_i(q)\,\tilde{x}_i(q)\,dq = \int_0^1 \bar{\phi}_i(q)\,x_i^*(q)\,dq. \tag{32}$$

Combining Equations (30) and (32) and expanding both sides as in Equation (27) yields

$$\int_0^1 \left(v_i - \phi_i(q)\right) \tilde{x}_i(q)\,dq = \int_0^1 \left(v_i - \bar{\phi}_i(q)\right) x_i^*(q)\,dq. \tag{33}$$

It remains to argue that the interim rule $\tilde{x}_i(\cdot)$ can be implemented by (at most) two posted prices. If $q_i^*$ does not lie in an ironed interval, then $\tilde{x}_i$ already corresponds to posting a single price $\hat{c}_i$. Otherwise, if $q_i^*$ in an ironed interval $[a, b]$, define the prices at the endpoints, $\hat{c}_{i,1} := F_i^{-1}(a)$ and $\hat{c}_{i,2} := F_i^{-1}(b)$ (so $\hat{c}_{i,1} \leq \hat{c}_{i,2}$). Notice that posting the larger price $\hat{c}_{i,2}$ with probability $\theta_i$ and the smaller price $\hat{c}_{i,1}$ with probability $1 - \theta_i$ implements $\tilde{x}_i$, wrapping up our claim. $\qquad\square$

We now define the ex-post budget feasible mechanism $\mathcal{M}_{\text{const}}^{\text{rand}}$. Note that for deterministic prices, we set $p_{i,1} = \hat{c}_i$, $\theta_{i,1} = 1$, and add a dummy option $p_{i,2} = 0$, $\theta_{i,2} = 0$, while for randomized prices we set $p_{i,1} = \hat{c}_{i,1}$, $\theta_{i,1} = \theta_i$, $p_{i,2} = \hat{c}_{i,2}$, $\theta_{i,2} = 1 - \theta_i$. Finally let $q_{i,j} = F_i(p_{i,j})$.

### C.4. Adapting proofs for the Generalized Utility Objective

This subsection records the modifications required to extend the arguments for the generalized utility in the Bayesian setting. The purpose is not to redo any derivations, but to make explicit which proof steps must be adjusted and why the adjustments are valid.

---

**Mechanism 4** Randomized Utility-Maximizing Ex-post Budget-Feasible Auction $\mathcal{M}_{\text{const}}^{\text{rand}}(\alpha, \beta; \mathbf{p}, \boldsymbol{\theta})$

---

1: Define $H = \{(i,j) : p_{i,j} \geq \frac{B}{\alpha}\}$ and $L = \{(i,j) : p_{i,j} < \frac{B}{\alpha}\}$. Let $U_{\text{price}}^H(\boldsymbol{p}, \boldsymbol{\theta}) = \sum_{(i,j) \in H} \theta_{i,j} (v_i - p_{i,j}) q_{i,j}$,
   $U_{\text{price}}^L(\boldsymbol{p}, \boldsymbol{\theta}) = \sum_{(i,j) \in L} \theta_{i,j} (v_i - p_{i,j}) q_{i,j}$, and $U_{\text{price}}(\boldsymbol{p}, \boldsymbol{\theta}) = U_{\text{price}}^H(\boldsymbol{p}, \boldsymbol{\theta}) + U_{\text{price}}^L(\boldsymbol{p}, \boldsymbol{\theta})$.
2: Independently draw a realized price $\widetilde{p}_i$ for each agent $i$ according to $\boldsymbol{p_i}, \boldsymbol{\theta_i}$. Correspondingly, partition the agents into
   $\widetilde{H} = \{i : \widetilde{p}_i \geq \frac{B}{\alpha}\}$ and $\widetilde{L} = \{i : \widetilde{p}_i < \frac{B}{\alpha}\}$.
3: **if** $U_{\text{price}}^H(\boldsymbol{p}, \boldsymbol{\theta}) \geq \left(1 - \frac{1}{\beta}\right) \cdot U_{\text{price}}(\boldsymbol{p}, \boldsymbol{\theta})$ **then**
4:     Order agents in $\widetilde{H}$ by decreasing $v_i - \widetilde{p}_i$. Iterate in this order and offer agent $i$ price $\widetilde{p}_i$ if the remaining budget is at
   least $\widetilde{p}_i$; otherwise, skip $i$ and continue.
5: **else**
6:     Order agents in $\widetilde{L}$ by decreasing $\frac{v_i - \widetilde{p}_i}{\widetilde{p}_i}$. Iterate in this order and offer agent $i$ price $\widetilde{p}_i$ if the remaining budget is at
   least $\widetilde{p}_i$; otherwise, skip $i$ and continue.
7: **end if**

---

**Lagrangian reduction and the posted-price structure** We need to verify that the Lagrangian relaxation continues to yield a separable optimization problem, and that each single-agent problem continues to admit an optimal threshold rule. The new Lagrangian objective is

$$\max_x \; \lambda B \; + \; \sum_i \mathbb{E}_{\mathbf{c}} \left[ \left( a_i v_i - (b_i + \lambda) \phi_i(c_i) \right) x_i(\mathbf{c}) \right].$$

For any fixed $\lambda$, the objective separates across agents and cost realizations and is optimized pointwise. In particular, for each agent $i$, the maximizer selects

$$\hat{c}_i \; = \; \phi_i^{-1}\left( \frac{a_i v_i}{b_i + \lambda} \right)$$

**Ironing: feasibility comparison and preservation on ironed intervals** We need to verify that ironing carries over under the generalized objective. Since Lemma 10 still holds and coefficients $a_i, b_i$ are non-negative, the feasibility and optimality properties between the ironed and non-ironed programs remain. As such we need to argue that the transformation from solution $x_i^*$ to $\tilde{x}_i$ retains its objective guarantee and its expenditure. The crucial note here is that under the generalized objective, coefficients $a_i, b_i$ appear as constant multipliers of expectations in the objective, and by linearity of expectation, all previously proven equalities still hold.

**Ex-post mechanism: definition changes under generalized utility** We now define the modified ex-post mechanism $\mathcal{M}_{\text{modified}}$:

---

**Mechanism 5** General-Utility-Maximizing Ex-post Budget-Feasible Auction $\mathcal{M}_{\text{modified}}(\alpha, \beta; \hat{\mathbf{p}})$

---

1: Partition the agents into $H = \{i : p_i \geq \frac{B}{\alpha}\}$ and $L = \{i : p_i < \frac{B}{\alpha}\}$, denote their respective ex-ante posted pricing
   generalized utility $U_{\text{price}}^H(\boldsymbol{p}) = \sum_{i \in H} (a_i v_i - b_i p_i) q_i$ and $U_{\text{price}}^L(\boldsymbol{p}) = \sum_{i \in L} (a_i v_i - b_i p_i) q_i$, and the total ex-ante
   posted pricing generalized utility as $U_{\text{price}}(\boldsymbol{p}) = \sum_{i \in L \cup H} (a_i v_i - b_i p_i) q_i$.
2: **if** $U_{\text{price}}^H(\boldsymbol{p}) \geq \left(1 - \frac{1}{\beta}\right) \cdot U_{\text{price}}(\boldsymbol{p})$ **then**
3:     Order agents in $H$ by decreasing $a_i v_i - b_i p_i$. Iterate in this order and offer agent $i$ price $p_i$ if the remaining budget is
   at least $p_i$; otherwise, skip $i$ and continue.
4: **else**
5:     Order agents in $L$ by decreasing $\frac{a_i v_i - b_i p_i}{p_i}$. Iterate in this order and offer agent $i$ price $p_i$ if the remaining budget is at
   least $p_i$; otherwise, skip $i$ and continue.
6: **end if**

---

The approximation ratio of this mechanism is the same as the one proved in Theorem 8. In the case analysis of the theorem, Case 1 directly follows by setting $u_i = a_i v_i - b_i p_i$. For Case 2, we need to decompose for the generalized utility objective similarly to Equations (20) and (21). We now give the analogous decompositions for the generalized objective. For notational

convenience, define the generalized utility per payment ratios (assumed to be decreasing)

$$\rho_i = \frac{a_i v_i - b_i p_i}{p_i}, \qquad \text{and set } \rho_{n+1} := 0.$$

The optimal independent fractional knapsack benchmark $U_B(\boldsymbol{p})$ admits the same telescoping decomposition:

$$
\begin{aligned}
U_B(\boldsymbol{p}) &= \mathbb{E}_{S \sim ind(\boldsymbol{q})}\big[u_B(S)\big] \\
&= \mathbb{E}_{S \sim ind(\boldsymbol{q})}\left[\sum_{i \in N} \rho_i \cdot \left(\min\Big\{B, \sum_{j \in S \cap \{1,\dots,i\}} p_j\Big\} - \min\Big\{B, \sum_{j \in S \cap \{1,\dots,i-1\}} p_j\Big\}\right)\right] \\
&= \sum_{i \in N} \Big(\rho_i - \rho_{i+1}\Big) \cdot \mathbb{E}_{S \sim ind(\boldsymbol{q})}\left[\min\Big(B, \sum_{j \in S \cap \{1,\dots,i\}} p_j\Big)\right].
\end{aligned}
\tag{34}
$$

The last equality follows by rearranging into a telescoping sum, using linearity of expectation, and the convention $\rho_{n+1} = 0$.

Likewise, the ex-ante independent posted-pricing benchmark $U_{\text{price}}(\boldsymbol{p})$ rewrites as

$$
\begin{aligned}
U_{\text{price}}(\boldsymbol{p}) &= \mathbb{E}_{S \sim ind(\boldsymbol{q})}\left[\sum_{i \in S} a_i v_i - b_i p_i\right] \\
&= \mathbb{E}_{S \sim ind(\boldsymbol{q})}\left[\sum_{i \in S} p_i \cdot \rho_i\right] \\
&= \mathbb{E}_{S \sim ind(\boldsymbol{q})}\left[\sum_{i \in N} \rho_i \cdot \left(\sum_{j \in S \cap \{i\}} p_j\right)\right] \\
&= \sum_{i \in N} \Big(\rho_i - \rho_{i+1}\Big) \cdot \mathbb{E}_{S \sim ind(\boldsymbol{q})}\left[\sum_{j \in S \cap \{1,\dots,i\}} p_j\right].
\end{aligned}
\tag{35}
$$

Comparing Equations (34) and (35), we see that both expressions have the same coefficient sequence $(\rho_i - \rho_{i+1})$ multiplying the same two cumulative expenditure terms; hence Lemma 9 applies unchanged to relate the two expectations.

# D. Inapproximability & Lower Bounds

## D.1. Welfare Prior-Free Lower Bound

In this subsection we prove that inapproximability results for the value objective naturally extend to the welfare objective as well. This implies a $\frac{1}{1+\sqrt{2}}$ lower bound for deterministic mechanisms for prior-free welfare maximization (based on the value maximization lower bound result from (Chen et al., 2011)), implying a small gap between our $\frac{1}{2+\sqrt{2}}$-approximation and the best-possible mechanism.

**Theorem 4.** *Let $\beta \in (0,1)$. If no budget-feasible mechanism can achieve a better than $\beta$-approximation to the value objective, then no budget-feasible mechanism can achieve a better than $\beta$-approximation to the welfare objective.*

*Proof.* Assume by contradiction there exists a DSIC-IR budget-feasible mechanism that gives a $\rho > \beta$ approximation to the welfare objective for some $\rho \in (0,1)$. We will use this mechanism to construct a $\rho' = \frac{\rho + \beta}{2} > \beta$ approximation DSIC-IR budget-feasible mechanism to the value objective.

Let $\delta = \frac{\rho - \beta}{2}$. Consider an instance $I = (\{v_i\}_{i \in [n]}, \{c_i\}_{i \in [n]}, B)$ for the value objective. We transform this instance to an instance for the welfare objective $I' = (\{v'_i\}_{i \in [n]}, \{c_i\}_{i \in [n]}, B)$, where $v'_i = v_i \cdot \Lambda$ for $\Lambda = \frac{B}{\delta \cdot v_{min}}$ (without loss of generality we may assume that $v_{min} \geq 0$). Consider the outcome of a $\rho$-approximate budget-feasible mechanism for the welfare objective on instance $I'$. Since $I'$ has the same sellers' costs as the original instance and the same budget constraint, this mechanism is DSIC-IR and budget feasible for the original instance $I$.

Let $S$ be the set of agents selected by running the mechanism on $I'$ and let $T$ be the optimal set of agents for the value objective for the original instance $I$. Since $S$ $\rho$-approximates the welfare objective, we have that for every other set $S'$ that satisfies $\sum_{i \in S'} c_i \leq B$, the welfare of agents in $S$ for instance $I'$ $\rho$-approximates the welfare of $S'$. Specifically, for set $T$, we have that

$$\sum_{i \in S}(v'_i - c_i) = \sum_{i \in S}(\Lambda \cdot v_i - c_i) \geq \rho \cdot \sum_{i \in T}(v'_i - c_i) = \rho \cdot \sum_{i \in T}(\Lambda \cdot v_i - c_i).$$

Rearranging gives that

$$\Lambda \cdot \sum_{i \in S} v_i \geq \rho \Lambda \cdot \sum_{i \in T} v_i - \rho \sum_{i \in T} c_i + \sum_{i \in S} c_i \geq \rho \Lambda \cdot \sum_{i \in T} v_i - \rho \cdot B > \rho \Lambda \cdot \sum_{i \in T} v_i - B,$$

where the second inequality follows budget feasibility. Dividing by $\Lambda$ gives

$$
\begin{aligned}
\sum_{i \in S} v_i &\geq \quad \rho \cdot \sum_{i \in T} v_i - \frac{B}{\Lambda} = \rho \cdot \sum_{i \in T} v_i - \delta \cdot v_{min} \\
&\geq \quad \rho \cdot \sum_{i \in T} v_i - \frac{\rho - \beta}{2} \cdot \sum_{i \in T} v_i = \frac{\rho + \beta}{2} \cdot \sum_{i \in T} v_i \\
&= \quad \rho' \cdot \sum_{i \in T} v_i,
\end{aligned}
$$

as desired, a contradiction to the assumption that no budget-feasible mechanism can obtain a better than $\beta$-approximation to the value objective. $\qquad \square$

**Corollary 5.** *No deterministic budget-feasible mechanism for welfare maximization can achieve an approximation ratio better than $\frac{1}{1+\sqrt{2}}$.*

### D.2. Bayesian Lower Bounds

We next provide two simple symmetric instances that clarify how tight our analysis is for the framework that converts ex-ante posted prices into a sequential posted-price mechanism. In these instances, the posted prices will be identical across agents, meaning that the behavior of the sequential posted-price mechanism depends only on the set of accepting agents and not on the ordering rule; thus, any ordering yields the same expected utility against the ex-ante posted-pricing benchmark. For the first instance (high prices), we prove that our analysis is tight at the $1 - \frac{1}{e}$ factor, matching the guarantees of Lemma 3. The second instance (low prices) shows that the $1 - O(1/\sqrt{k})$ approximation guarantee in Lemma 2 is tight up to constants.

**Lemma 11.** *There exist instances with ex-ante optimal prices $\hat{c}$ that are individually ex-post budget feasible ($\hat{c}_i \leq B$), for which the best approximation to the ex-ante posted pricing utility $U_{price}(\hat{c})$ achievable by any sequential posted-pricing mechanism using these prices converges to $1 - \frac{1}{e}$.*

*Proof.* Fix $n$ agents and consider the symmetric instance

$$B = \frac{1}{n}, \qquad v_i = 2B, \qquad c_i \sim U[0,1] \quad \text{for all } i \in [n].$$

For this instance, the ex-ante optimal prices $\hat{c}_i$ and quantiles $q_i$ are $\hat{c}_i = q_i = B$ while the ex-ante posted pricing utility benchmark is

$$U_{\text{price}}(\hat{c}) = \sum_{i=1}^{n}(v_i - \hat{c}_i)q_i = n \cdot B \cdot \frac{1}{n} = B.$$

Now consider any sequential posted-pricing mechanism that uses these prices. Notice that since everything in this instance is symmetric (values, distributions, prices) no ordering rule can differentiate between agents. Additionally, since each posted price equals the full budget, once an agent accepts, the mechanism can serve no further agents. Therefore for any sequential posted-pricing mechanism the realized utility $U^{\text{seq}}(\hat{c})$ is $B$ if at least one agent can accept and 0 otherwise, so

$$U^{\text{seq}}(\hat{c}) = B \cdot \Pr[\exists \text{ at least one agent that can accept}] = U_{\text{price}}(\hat{c}) \cdot \left(1 - \left(1 - \frac{1}{n}\right)^n\right).$$

Since

$$1 - \left(1 - \frac{1}{n}\right)^n \to 1 - \frac{1}{e} \qquad \text{as } n \to \infty,$$

the approximation ratio of any sequential posted-pricing mechanism using these prices converges to $1 - \frac{1}{e}$, proving the claim. $\qquad \square$

**Lemma 12.** *For any integer $k \geq 1$, there exist instances with ex-ante optimal prices $\hat{c}$ that are individually ex-post budget feasible ($\hat{c}_i \leq B$), for which the best approximation to the ex-ante posted pricing utility $U_{price}(\hat{c})$ achievable by any sequential posted-pricing mechanism using these prices is at most $1 - \frac{1}{4\sqrt{k}}$.*

*Proof.* Fix $n = 2k$ agents and consider the symmetric instance

$$v_i = \frac{2B}{k}, \qquad c_i \sim U\left[0, \frac{2B}{k}\right] \qquad \text{for all } i \in [n].$$

For this instance, we can compute the ex-ante optimal prices $\hat{c}_i = \frac{B}{k}$ and the ex-ante probability $q_i = \frac{1}{2}$, while the ex-ante posted pricing utility benchmark is

$$U_{\text{price}}(\hat{c}) = \sum_{i=1}^{2k}(v_i - \hat{c}_i)q_i = 2k \cdot \frac{B}{k} \cdot \frac{1}{2} = B.$$

Notice again that any sequential posted-pricing mechanism that uses these prices cannot differentiate between agents (due to symmetry). Under the corresponding sequential posted-pricing mechanism, each accepting agent contributes utility $v_i - \hat{c}_i = B/k$, and the mechanism can serve at most $k$ agents. Let $X \sim \text{Bin}(2k, 1/2)$ and $U^{\text{seq}}(\hat{c})$ the expected utility of any sequential posted-pricing mechanism. Then

$$U^{\text{seq}}(\hat{c}) = \frac{B}{k}\,\mathbb{E}_X[\min(X, k)] = U_{\text{price}}(\hat{c})\frac{\mathbb{E}_X[\min(X, k)]}{k}.$$

We can rewrite $\min(X, k) = k - (X - k)^+$, and simplify $\mathbb{E}[(X - k)^+] = \frac{1}{2}\,\mathbb{E}[|X - k|]$ (because $X$ is symmetric around $k$) and $E[|X - k|] = k\Pr[X = k] = k \cdot \frac{\binom{2k}{k}}{2^{2k}}$ (by the definition of $X$). Therefore,

$$\frac{\mathbb{E}[\min(X, k)]}{k} = 1 - \frac{\mathbb{E}[(X - k)^+]}{k} = 1 - \frac{1}{2}\Pr[X = k] = 1 - \frac{\binom{2k}{k}}{2^{2k+1}}.$$

Finally, using a standard bound (we can prove this from Stirling's inequality)

$$\binom{2k}{k} \geq \frac{2^{2k}}{2\sqrt{k}},$$

we can prove our claim

$$\frac{U^{\text{seq}}(\hat{c})}{U_{\text{price}}(\hat{c})} = 1 - \frac{\binom{2k}{k}}{2^{2k+1}} \leq 1 - \frac{1}{4\sqrt{k}}.$$

$\qquad \square$

# E. Computational Considerations

**Runtime of Mechanisms:** The welfare-approximate Mechanism 1 is extremely efficient, and can run in time $O(n \log n)$, since sorting $\frac{v_i}{c_i}$ is the only non-linear step. Similarly, the utility-approximate Mechanism 2 is efficient and runs in time $O(n \log n)$, with sorting (either $\frac{v_i - p_i}{p_i}$, or $v_i - p_i$) being the only non-linear step.

**Computing Ex-ante Prices:** Following the discussion in Appendix C.1, under *regularity*, the optimal prices are easy to compute: for any fixed $\lambda$, each $\hat{c}_i$ is obtained independently as $\phi_i^{-1}(v_i/(\lambda+1))$, so the Lagrangian program decomposes across agents and can be solved efficiently. Since the total expected payment is monotone in $\lambda$, the budget-binding multiplier can then be found by a simple binary search over $\lambda$, and hence the entire computation is efficient.

Cases with *irregular* distributions require slightly more computation, but overall remain efficient. The ironing process needs to be performed only once, after which the ironed virtual function is used in the previously described approach (for regular distributions). The computational burden of ironing amounts to computing a concave hull. In the discrete finite-support setting there exists an $O(\bar{v}\log\bar{v})$ construction (Elkind, 2007) (where $\bar{v}$ is the maximum value of the distribution). In the continuous setting, we can choose a $\varepsilon$-discretization of the distribution (in $O(\bar{v}/\varepsilon)$ time) and then follow the same process as for discrete distributions.

