# OpenReview forum: "From Welfare to Utility: Generalized Objectives in Budget-Feasible Procurement"
_ICML.cc/2026/Conference — ICML 2026 regular_

### Official Review · Reviewer_QFHr · 2026-03-12

**Soundness:** 3
**Presentation:** 3
**Significance:** 3
**Originality:** 3
**Overall Recommendation:** 4
**Confidence:** 3

**Summary:**

The paper studies the budget-feasible procurement problem, relevant in any scenario where a buyer seeks to procure goods or services from self-interested sellers and is constrained by a budget. The authors give some examples in the modern computing ecosystem where such a setting may arise. In the model of this paper, there is a single buyer with additive utility and $n$ sellers. A mechanism takes the reported production costs of sellers and produces an allocation and a payment. The goal is to construct DSIC-IR mechanisms that maximize the social welfare or the buyer utility. Note that due to their great relevance, both the utility and welfare objectives have been studied in the procurement auction setting already, only without budget constraints. The authors also look at generalizations of these objectives. The mechanism has either no (prior-free setting) or distributional information (stochastic setting) and it is compared against a benchmark with full information of the costs of the sellers.

The procurement problem is quite well-studied due to its practical significance, but the vast majority of work focuses only on the objective of value maximization. The justification is that the buyer has already set aside the budget, and thus it can be consumed in full. However, the paper argues that in many settings, unused budget could be reused elsewhere, making buyer utility (that is, the buyer's total value minus the payments) a more appropriate objective.

The main results of the paper are the following:

For the social welfare objective, the authors consider the prior-free setting, in which we maximize our objective over worst-case inputs of seller costs. They provide a simple $1/(2+\sqrt{2})$-approximation.

Since no guarantee for buyer utility is possible in the prior-free setting, they investigate the Bayesian setting (that is, sellers’ costs are now drawn from a known distribution). First, they ​​solve the optimization problem optimally when the buyer’s budget constraint is satisfied in expectation. Then, building on that, they design a simple posted pricing mechanism that guarantees a $⅕$ -approximation for the case that the budget constraints are satisfied ex-post.

Note that both mechanisms are polytime computable. Several extensions on different directions also presented, such as one that states that the results for the utility objective extend to any additively separable affine objective with non-negative weights.

On the technical side, they extend and improve upon the work of Balkanski & Hartline by 1) removing the $k$-large market assumption and 2) fixing a small hole in their analysis.

**Compliance With Llm Reviewing Policy:**

Affirmed.

**Final Justification:**

I appreciate the authors' detailed responses, which resolved most of my questions. I, thus, updated my score to weak accept, meaning I would be fine for the paper to be accepted if someone is very positive about it.

**Key Questions For Authors:**

Are there any lower bounds for the proposed mechanisms, or do you know of any that transfer to this setting?

Why do you study only DSIC-IR mechanisms and don’t consider at least other weaker notions of truthfulness? Do we expect in real life for such mechanisms to be, or intended to be, DSIC?

A prior-free setting is probably always too pessimistic. Why didn’t you also consider some stochastic setting for welfare? And then you go directly to full knowledge of the distribution when prior-free has no guarantees, which is also strong. If you were to have only samples from the distribution, is there a straightforward way to argue that a similar mechanism could also provide some guarantees there?

Can you expand a bit more on the motivation, with some specific scenario that such a mechanism has been implemented or you see that this mechanism can be implemented? Maybe you can go into more detail in one of those that you mention in the introduction.

**Limitations:**

yes

**Strengths And Weaknesses:**

**Strengths**

The part of the motivation that some budget can be used elsewhere, to motivate the study of utility, is good. The budget constraints are also natural, making the model of the paper a natural consideration given previous work on procurement auctions.

The paper is well-written. The presentation is solid and the results and proofs are easy to follow. The comparison with previous work is there, making the contributions and technical novelties of the paper clear.

For the utility objective, the constant-factor approximation with a reasonably large constant is a positive result and it comes with some careful technical steps to arrive there. They start with the ex-ante case, where they provide an optimal result based on the observation that it is without loss to restrict attention to utility maximization in instances in which the budget constraint is tight. Then they extend the posted prices to the ex-post case, where they partly use and correct the previous approach of Balkanski & Hartline, which is independently a contribution of its own.

**Weaknesses**

I’m not super convinced about the motivations given in the introduction, as these examples of where such a mechanism could be applied don’t provide much detail on the actual implementation. How is it the case in these examples that all sellers report their production costs (I assume they are normally kept private in these cases) and the buyer values each service at a known $v_i$? (see also similar question in the next section).

If we are not convinced that we actually need to study this model and have a good answer for this specific case of procurement auctions, then the theoretical contribution is not enormous, and the paper overall doesn’t have a strong point of focus, either in theory or in practice. The paper goes into two objectives and various extensions that are all somewhat motivated, but on the technical front it doesn’t seem to offer much new beyond specific techniques to this setting.
Continuing on this, I understand that it’s a theory paper, but given that the techniques are quite specific to this problem and there are not broader lessons learnt on the theory side, it could benefit from experimental results to show if these mechanisms that are poly-time computable in theory can actually be applied and at what scale, and also if experimentally they would do much better than their theoretical bounds, which would strengthen their applicability, since, for instance, the prior-free setting, is for sure overly pessimistic. Some experiments would also strengthen the case that some of the examples mentioned in the intro, can be actually tackled in practice with such a budget-feasible procurement problem.

There are no lower bounds for either of the objectives. Although the constants in the constant-factor approximations are quite good, we still don’t have much indication if this is actually the type of mechanism we’d safely choose to implement.

Some small typos for the authors to fix:

P.2: “to optimal the social welfare…”, “parallels that the approach…”
P.5: “contribute to much to the…”
P.7: “to the Appendix Appendix C.2…”

---

> ### Author Rebuttal · Authors · 2026-03-30
>
> > How do sellers report their private costs and why are buyer values known?
>
> The model we use (that a buyer knows their value for each service, that seller costs are private) has a long and well-established history in both economics and algorithmic game theory, and has been adopted in practice precisely because it captures real-world settings accurately. Specifically, when a firm purchases services, it has an accurate estimate of the value these services can generate for the firm. Moreover, it is unrealistic to assume the firm knows the private costs of the suppliers, as these are typically not reported. These private values need to be *elicited*, which is why the mechanism must consider incentives in order to incentivize truthful reporting. The Revelation Principle shows it is equivalent to consider incentive-compatible mechanism design and non-truthful mechanism design with equilibrium analysis with respect to the objective. Regarding the practicality of the model, we note that research teams at major technology companies (e.g., Google, Meta, Amazon, and Microsoft) have adopted mechanisms grounded in exactly these models for designing and running their procurement and resource-allocation systems at scale. The model is not an abstraction for abstraction's sake; it is the abstraction that practitioners use. (See, e.g., Anari et al. FOCS14, OR18, Bei et al. STOC12, Bhawalkar et al. EC25, Chen et al. SODA11, Deng et al. ICML25.)
>
> > Q1: Lower bounds
>
> Thanks for pointing this out. See the lengthy discussion in the response to reviewer htT4, inspired by the important points raised in your reviews.
>
> > Q2: DSIC-IR
>
> We focus on DSIC-IR mechanisms because they provide the strongest and most transparent incentives: truthful reporting is the best response regardless of others' behavior, making this strategy straightforward and easy to follow. In contrast, weaker notions require agents to reason about other agents' beliefs, strategies, and type distributions, which is more demanding than having a simple dominant strategy of simply reporting your type, regardless of other agents' reports. Moreover, our mechanisms for the Bayesian case are *extremely* simple: the buyer just sets a fixed take-it-or-leave-it price for each seller, who accepts the price if it's higher than their production cost. These posted-price mechanisms are widely deployed in practice due to their implementation and participation simplicity.
>
> > Q3: Prior-free vs stochastic vs samples
>
> We agree that the prior-free setting is pessimistic—it is, by definition, worst-case. However, obtaining a positive result in this setting for welfare then only shows how strong this result is, obtaining a 0.29-approximation even in the worst-case. This 0.29-approximation then immediately extends to any setting with more information (samples, Bayesian, etc). It is also aligned with the Nobel-prize-winning Wilson doctrine, lauded by economic practitioners for postulating that mechanism designers should offer solutions that do not depend on market details (such as distributions) because they may be unknown to practitioners or are subject to intractable change. It is also still useful to characterize that without any additional information, utility maximization is hopeless (Prop. 4), and hence we move to the Bayesian (or stochastic, average-case) setting for utility maximization.
>
> For the utility objective, known distributions allow us to characterize the optimal ex-ante mechanism and design near-optimal ex-post mechanisms. The two questions of (1) how many samples are required to effectively learn the sellers' cost distributions to approximately implement this optimization and (2) can one effectively use samples to run a mechanism without learning the distributions first are fantastic questions for future work. Both of these directions have been tackled for other areas of mechanism design (e.g., revenue maximization, see for example the introduction of [Guo et al. STOC19]), but really require the foundations of solving the distributional problem first to form a benchmark so one can best understand how sample-complexity bounds trade off with approximations and fit into the bigger picture. Our work contributes this first foundational Bayesian result, and now future work will be able to attempt to answer sample questions.
>
> > Suggestion for experiments
>
> Thanks for your suggestions, we would be happy to add simulations to demonstrate how our mechanisms might perform in practice. Note however, that simulations will be, by definition, average-case, so they are incomparable with the prior-free setting and will align with the Bayesian (average-case) setting.
>
> > Q4: Mechanism motivation, example
>
> Our mechanism is from the simplest possible class that is widely deployed in practice. Concretely, we direct the reviewer to the model of pricing LLM APIs introduced by Google Research (Bhawalkar et al. EC25) for a very realistic scenario where such mechanisms can be deployed.

---

> > ### Author Rebuttal · Reviewer_QFHr · 2026-04-04
> >
> > Thank you for your detailed response. I'm willing to update my score, given the replies, in particular the updates on the lower bounds. Since there is no experimental section, I would still appreciate it if you could briefly expand on the pricing scenario of the LLM APIs. Are there any assumptions of your theoretical framework that could break down in practice, and if yes, would the core ideas of the mechanisms be easily adjusted to retain the good approximation guarantees? And are there any challenges, e.g., on the computational front, that you foresee if these mechanisms were to be deployed in such large-scale instances?

---

> > > ### Author Response · Authors · 2026-04-07
> > >
> > > We thank the reviewer for taking into consideration the ICML length restrictions and giving us a chance to expand.
> > >
> > > > I would still appreciate it if you could briefly expand on the pricing scenario of the LLM APIs.
> > >
> > > Regarding the specific LLM API application brought up by the Google Research team, they consider a model where a generic LLM (the buyer) gets a prompt from a user and a budget constraint. Then, the generic LLM produces an answer by prompting APIs of specialized LLMs (the sellers), and synthesizing their results into a high-quality answer for the user. The specialized LLMs have private costs, and are getting compensated for their services, subject to the buyer’s budget constraints. Unlike our implementation, Bhawalkar et al. consider an indirect mechanism where sellers report a price (rather than their private production cost), and very similarly to our direct mechanism for the prior-free case, then the mechanism takes sellers by descending value-over-price ratio until the budget constraint is met. Since the mechanism is not truthful, Bhawalkar et al. analyze the efficiency of the equilibrium of their mechanism. In this sense, in Bhawalkar et al., the sellers are price setters, while in our mechanisms, the sellers are price takers. In addition, in order to form the equilibrium, the sellers are required to know the value of the buyer for their services, an assumption we do not need to make.
> > >
> > > > Are there any assumptions of your theoretical framework that could break down in practice, and if yes, would the core ideas of the mechanisms be easily adjusted to retain the good approximation guarantees?
> > >
> > > We first note that every theoretical model, when implemented, must be adapted to the application at hand, with its specific idiosyncrasies. While it is hard to foresee all the additional challenges that will arise when implementing such mechanisms in practice, we believe the following might pop up:
> > > - While additive valuations are frequently seen in practice, there are also some scenarios where they do not capture a buyer’s value, i.e., where  a marginal value of procured goods might depend on other procured goods. In such cases, the mechanisms will need to be adapted. However, we believe that basic ideas used in our mechanisms, such as (marginal) value-over-cost ordering and pricing based on soft budget relaxation, will be useful in such settings as well.
> > > - In the LLM API motivation, introduced by Google, it is assumed the sellers have no production constraints. In some cases, specialized LLMs might have limited compute, which might introduce an additional layer of competition among buyers, which buyers need to take into account when setting prices. However, the way to model this would be in a multi-buyer setting, which in addition to a procurement setting, becomes a two-sided market with budgeted buyers. This is a fantastic problem for future work, but will introduce all of the additional complexities that come with two-sided markets.
> > >
> > > > And are there any challenges, e.g., on the computational front, that you foresee if these mechanisms were to be deployed in such large-scale instances?
> > >
> > > The prior-free mechanism is extremely efficient, and can run in time O(nlogn) (sorting is the only non-linear step). The Bayesian mechanism is highly efficient (again, sorting is the only non-linear step) and easy to run once the soft-budget component is solved. Solving the ex-ante program can be done efficiently, and in practice is linear time, as described below.
> > >
> > > Solving the ex-ante program: For regular distributions, this requires repeatedly solving the Lagrangian program for different values of $λ$. For any fixed $λ$, the threshold rule from the proof of Lemma 8 implies that the Lagrangian program can be solved in $O(n)$ time. Let $\bar v = \max_i v_i$ be an upper bound on any value for any service (so in practice, at most a trillion). A binary search over $λ$ is then used to find the optimal value in time $O(\log{\bar v})$. Hence in total, the running time is $O(n \log{\bar v})$ (or for $\bar v$ at most a trillion, $O(n)$). Irregular distributions require slightly more computation, but overall are still efficient, and the extra computation can be preprocessed and amortized over time.

---

### Official Review · Reviewer_htT4 · 2026-03-13

**Soundness:** 3
**Presentation:** 3
**Significance:** 2
**Originality:** 2
**Overall Recommendation:** 4
**Confidence:** 4

**Summary:**

This paper studies budget-feasible procurement under standard additive values, and shifts the objective from the traditional value-maximization benchmark to welfare, utility, and related generalized objectives. The paper develops a prior-free approximation for welfare, proves an impossibility result for prior-free utility, and then turns to the Bayesian setting to derive an optimal soft-budget posted-price characterization and a constant-factor approximation for the hard-budget case. The work is technically careful, uses a standard and reasonable model, and also corrects an issue in the analysis of earlier literature.

**Compliance With Llm Reviewing Policy:**

Affirmed.

**Final Justification:**

I am positive on this paper. I will keep my score as it is.

**Key Questions For Authors:**

1. Can the authors better justify the practical relevance of the Bayesian budget models, especially the interpretation of ex-ante budget feasibility in realistic procurement environments?
2. Are there any potential results that would clarify how the current approximation guarantees are close to optimal. Furthermore, to what extent are the current constant factors an artifact of the proof technique rather than an inherent barrier of the problem?
3. Which parts of the analysis genuinely require the present mechanism design template, and which parts might admit stronger guarantees through a different mechanism or a refined argument?

**Strengths And Weaknesses:**

Strengths
1. The model is standard, clean, and well motivated within the budget-feasible procurement literature. The assumptions appear reasonable for the theoretical setting considered.
2. The paper is technically solid and unusually complete: it provides positive results, negative results, and extensions to irregular distributions and more general objectives.
3. The derivations are careful, and the overall solution path is coherent. In particular, the Bayesian analysis is developed in a systematic way from soft-budget optimality to hard-budget approximation.
4. The paper identifies and fixes a subtle issue in the earlier analysis of related prior work, which is a meaningful technical service to the literature.
Weaknesses
1. The main technical ideas are substantially parallel to prior literature. Much of the contribution reads as a careful extension and completion of known lines of argument rather than a strongly new conceptual breakthrough.
2. The practical relevance of the Bayesian budget assumptions is not fully convincing. In particular, the real-world interpretation of ex-ante budget feasibility is debatable in many procurement settings.
3. The paper does not provide tight lower bounds, so it is difficult to judge how close the current approximation factors are to the best possible guarantees. It seems plausible that stronger guarantees should be obtained by modifying the approximation mechanism or refining the analysis, given that the present methods may be approaching their natural limits; the paper would benefit from a clearer discussion of which barriers are technical and which are likely fundamental.

---

> ### Author Rebuttal · Authors · 2026-03-30
>
> > Q1: Bayesian and ex-ante budget feasibility
>
> Prop. 4 shows that for utility maximization, additional structure is necessary. In practice, this structure comes from data; procurement is often repeated with similar suppliers, so the buyer can estimate the sellers' distributions from historical observations, justifying a Bayesian assumption. Bayesian models are standard and widespread in the economic literature as optimizing over past data generates better outcomes.
>
> We primarily care about ex-post budget feasibility, as it guarantees that the budget is never exceeded. Ex-ante budget feasibility is mainly a tractable analytical relaxation used en route, used to design ex-post budget feasible mechanisms, which enforce the operational constraint we ultimately care about.
>
> We will add a discussion on ex-ante vs. ex-post feasibility, including an example where the optimal ex-ante mechanism violates the budget constraint ex-post on every non-zero allocation, as well as ex-ante motivation from repeated settings where overspending averages over time, to the final version of the paper.
>
> > Q2+3: Tightness and lower bounds
>
> We thank the reviewers for these questions and intend to introduce an in-depth discussion and new detailed proofs in the final version.
>
> *Welfare*: inspired by the reviewers' questions, we prove that for any $β\in(0,1)$, a budget-feasible mechanism that gets a $ρ>β$ approximation to welfare can be transformed into a budget-feasible $ρ'>β$-approximation to value. This implies that any lower bound for the widely-studied value objective automatically transfers to the welfare objective. Specifically, the $1/(1+\sqrt{2})$ lower bound for deterministic mechanisms devised by Chen et al. '11 transfers to welfare as well, implying a small, but existing, gap between our $1/(2+\sqrt{2})$-approximation and the best-possible mechanism. We note that closing such gaps is challenging at times, as the gap for the well-studied value objective has not yet been closed after 16 years.
>
> The reduction works as follows: given an instance $I=(\vec v,\vec c,B)$ for the value objective, we transform $I$ into an instance $I'=(\vec v',\vec c,B)$ for the welfare objective, where for all $i\in[n]$, $v_i'=Λ\cdot v_i$ for $Λ=\frac{B}{δv_{\min}}$, where $δ=(ρ-β)/2$ and $v_{\min}$ is the minimal seller value. As the costs and budget haven't changed in $I'$, DSIC-IR and budget-feasibility follow from the properties of the $ρ$-approximate mechanism. To prove the improved approximation, we consider $S$, the allocation obtained by the mechanism, and $T$, the optimal allocation for the value objective. Since the mechanism is $ρ$-approximate to welfare, we have that $$\sum_{i\in S}v_i'-c_i\ge ρ(\sum_{i\in T}v_i'-c_i).$$ Using basic arithmetic manipulations, this implies that $$Λ \sum_{i\in S} v_i\ge ρΛ\sum_{i\in T}v_i-B.$$ Dividing both sides by $Λ$ gives that $$\sum_{i\in S}v_i\ge ρ\sum_{i\in T}v_i- δ\cdot v_{\min}\ge ρ\sum_{i\in T}v_i-\frac{ρ-β}{2}\cdot \sum_{i\in T}v_i=\frac{ρ+β}{2}\cdot\sum_{i\in T}v_i,$$ implying a better than $β$ approximation to the value objective.
>
> *Utility*: the question pertains to the ex-post results, where the objective is the best truthful mechanism, and lower bounds are computational. Finding the optimal mechanism is NP-hard with ex-post feasibility, as with point-mass cost distributions, as solving this problem requires solving the knapsack problem optimally. The question is then whether we can show computational hardness of approximation. These results are sparse in the mechanism design literature (just Collina Weinberg EC20 and Cai et al. JACM26), and we are not aware of any such results for procurement. We believe this question is a great question for follow-up works, and beyond the scope of our paper.
>
> We now show barriers to improving the analyses in Lemmas 2 and 3.
>
> We give a tight instance for the analysis of Lemma 3 when $α=1$: with budget $B=1/n$ and $n$ symmetric agents with value $v_i=2B$ and costs $c_i\sim U[0,1]$. The ex-ante posted price for every agent is $p_i=B$, and each agent accepts with ex-ante probability $1/n$, so the ex-ante total acceptance probability is exactly 1. Thus the optimal ex-ante utility is exactly $B$. Since agents are symmetric, all ex-post feasible posted price mechanisms provide the same expected utility, giving expected utility $B\cdot\Pr[\exists\text{agent that accepts the price}] = B\cdot(1-(1-\frac{1}{n})^n)\approx B(1-1/e)$, yielding a tight lower bound of $(1-1/e)$.
>
> Similarly, for Lemma 2, we construct an instance with $n=2k$ symmetric agents with values $2B/k$ and costs in $U[0,2B/k]$. The ex-ante optimal prices are to $B/k$ and by symmetry, any sequential posted-price mechanism using these prices yields the same expected utility. The ratio to the ex-ante benchmark is then $\frac{\mathbb{E}[\min(X,k)]}{k}$ where $X\sim Bin(2k,1/2)$, and this is at most $1-\frac{1}{4\sqrt{k}}$. This shows a $Θ(1-1/\sqrt{k})$ dependence in Lemma 2 is unavoidable.

---

> > ### Author Rebuttal · Reviewer_htT4 · 2026-04-04
> >
> > Thank you for your detailed response. I appreciate the clarification and will keep my score as is.

---

### Official Review · Reviewer_pZj4 · 2026-03-13

**Soundness:** 3
**Presentation:** 3
**Significance:** 3
**Originality:** 3
**Overall Recommendation:** 4
**Confidence:** 4

**Summary:**

This paper studies mechanism design for the budget-feasible procurement problem. Unlike prior work that focuses on buyer value maximization, this paper considers the buyer as a utility maximizer (value minus payments). The authors provide a simple mechanism achieving a 1/(2+√2)-approximation to the optimal social welfare in the prior-free setting and show that the buyer's utility objective is inapproximable in the prior-free setting (Proposition 4). In the Bayesian setting, they first derive the optimal mechanism when the budget constraint holds in expectation (ex-ante), and then construct a posted pricing mechanism that guarantees a 1/5-approximation to the optimal utility under ex-post budget constraints. The paper also extends these results to more general objectives using ironing techniques.

**Compliance With Llm Reviewing Policy:**

Affirmed.

**Final Justification:**

No more questions, and I prefer weak accept.

**Key Questions For Authors:**

See Weakness

**Limitations:**

Yes

**Strengths And Weaknesses:**

**Strengths**

1. This paper gives the first simple mechanism that achieves a constant approximation of the buyer's utility while satisfying ex-post budget constraints.
2. The impossibility result (Proposition 4) for utility maximization in the prior-free setting is clean and provides a clear motivation for studying the Bayesian setting.
3. The structural characterization of the ex-ante optimal mechanism is elegant and serves as a useful stepping stone for the ex-post result.
4. The use of ironing to handle irregular cost distributions demonstrates generality and practical relevance.

**Weaknesses**
The assumption that the buyer is a utility maximizer (value minus payments) rather than a value maximizer subject to budget constraints deserves more discussion. In many procurement and online advertising settings, the buyer aims to maximize value subject to a budget, because the budget has already been allocated and cannot be reused elsewhere. The paper's motivation that "unused budget could be reused elsewhere"  implicitly assumes the procurement problem is embedded in a larger budget allocation context, but this is not formally modeled. A more compelling justification would be to explicitly model the outside option for unused budget, or to provide concrete applications in which utility maximization is clearly the appropriate objective.
2

---

> ### Author Rebuttal · Authors · 2026-03-30
>
> > Q1: “The assumption that the buyer is a utility maximizer (value minus payments) rather than a value maximizer subject to budget constraints deserves more discussion. In many procurement and online advertising settings, the buyer aims to maximize value subject to a budget, because the budget has already been allocated and cannot be reused elsewhere. The paper's motivation that "unused budget could be reused elsewhere" implicitly assumes the procurement problem is embedded in a larger budget allocation context, but this is not formally modeled. A more compelling justification would be to explicitly model the outside option for unused budget, or to provide concrete applications in which utility maximization is clearly the appropriate objective.”
>
> Thank you for this point. First, utility maximization is well-justified beyond reusing the budget. Consider a firm that wants to maximize its profit subject to liquidity constraints. The firm can purchase services that generate revenue (the values of the purchased goods), and has to compensate suppliers for its purchases, but is limited by its liquidity. Utility subject to budget constraints is analogous to the firm maximizing its net profit, which is a more meaningful metric than gross profit. We can add this justification to the paper. Second, when assuming the budget can be reused elsewhere, this objective is also well-motivated. For instance, Google allocates budgets to each of its teams and then runs an internal procurement auction for computing resources (GPU clusters, serving capacity, data pipeline priority, etc). Each team wants to maximize their utility subject to their budget, but whatever portion of the budget goes unused can be reallocated to other team priorities (e.g., hiring headcount, or being banked for the future). The same is true at Microsoft, Meta, and Amazon. Our work models the problem of an individual team once it has been allocated a budget, but knows it's worthwhile to tradeoff how much is being spent per value gained, as that budget can be reused, hence we do not model the larger question (and not part of the team's decision) of how to allocate that budget. It would, however, be interesting to, in follow-up work, study the much more complex dynamic setting of considering unspent budget. Note also that in the literature for budgeted agents in forward auctions, these budgeted buyers are always utility-maximizing.

---

> > ### Author Rebuttal · Reviewer_pZj4 · 2026-04-04
> >
> > Thank you for your clarify.

---

### Official Review · Reviewer_cVmV · 2026-03-14

**Soundness:** 3
**Presentation:** 3
**Significance:** 3
**Originality:** 2
**Overall Recommendation:** 3
**Confidence:** 4

**Summary:**

This paper studies budget-feasible procurement auctions: a budget constrained buyer wishes to buy goods/services from multiple sellers. Seller i provides value v_i to the buyer, but incurs a cost c_i of providing the service. The procurement mechanism solicits the costs from the sellers, posts payments p_i based on the reported costs which do not exceed the buyer budget B, and decides which goods to buy pays the selected sellers. Seller utility is p_i - c_i, and the sellers can misreport costs to gain more utility. We are interested in mechanisms that satisfy DSIC and IR.

While literature mostly focuses on maximizing buyer value, the paper considers the objectives of maximizing total welfare (buyer value minus seller costs) and buyer utility (value minus payments). In the prior free setting, they give a knapsack based DSIC-IR mechanism that achieves constant approximation to optimal welfare, and show that no approximation to the optimal utility is possible. Therefore they relax the setting from prior free to the Bayesian setting. They first give a utility optimal mechanism which only satisfies the budget constraint ex-ante, and later show that this can be adapted to satisfy the constraint ex-post while obtaining at least 0.2021 fraction of the optimum utility.

**Compliance With Llm Reviewing Policy:**

Affirmed.

**Final Justification:**

While the author rebuttal did not significantly change my opinion about the modeling choice of the utility function or the SW objective in this setting, I am not opposed to accepting the paper if other reviewers chose to support it. It would be good to add a discussion of the mentioned points to the final version.

**Key Questions For Authors:**

1. The paper motivates motivates utility maximization since unused budget is valuable to the buyer, and yet the formalization seems to discard the very budget surplus that motivates the objective. It seems like Proposition 4 is sensitive to the exact utility formalization, and the paper should explain more carefully why v−p is the economically correct notion rather than v+(B−p), or clarify that the impossibility is tied to this specific formulation, or construct a new example for the v+B-p formulation.

2. Although the welfare objective is mathematically natural and gives interesting theoretical results, its economic interpretation in the procurement setting is unclear, and the paper would benefit from a much clearer explanation of who the optimizing principal is supposed to be when welfare is the objective, and why that objective should arise endogenously in this market.

**Limitations:**

yes

**Strengths And Weaknesses:**

**Strengths:**

The paper is well motivated as it studies the newer objectives of utility in procurement auctions with a budget-constrained buyer. The theoretical contributions are interesting, the mechanisms are clean, and the application to generalized objectives is nice. The paper is also well-written.

**Weakness:**

In terms of techniques, it seems like the novelty is limited, since the techniques are adaptations of greedy mechanisms for budget-feasible value maximization, and the results in the Bayesian setting rely heavily on prior work. As for the model itself, I have two questions/criticisms:

First, the authors consider utilities motivated by the fact that "in many settings, unused budget could be reused elsewhere", implying that the buyer derives value from unspent money. In this case, why is the utility modelled as value - payment and not value + unspent budget, i.e., v - p instead of v + B - p? This may seem like a cosmetic issue since it's only an additive shift, but the modelling choice affects interpretation and more importantly, the results of the paper that pertain to approximation -- in particular, the negative result from Proposition 4 which shows that the optimial utility is inapproximable in the prior free setting. In their construction, if p<1, the mechanism gets v-p = 0 while the optimal v-p equals 1 - c; and if p = 1, then v-p = 0 while the optimal utility is 1- \eps. While this shows an unbounded gap for v-p, the gap between v+B-p is only 1/2: if p<1, the buyer has unspent budget of 1 in the mechanism, while the optimal v+B-p is 2-c; and if p=1, the buyer has unspent budget 1 and the optimal utility is close to 2-\eps. This suggests that appropriately defining the utility by incorporating unspent budget can prevent a strong negative result like Prop 4; at the very least it suggest that the example from Prop 4 doesn't work.

Second, while utility maximization seems like a natural objective, I found the case for welfare maximization less convincing in the current model. Welfare is the value minus sellers’ production costs, but in the procurement setting studied here the mechanism is run by the buyer, and a rational buyer would typically care about their own value or utility, not total welfare. The paper (somewhat weakly) motivates the welfare objective through settings in which the value of the buyer creates value for society and the seller costs harm the environment. But in such a setting a procurement auction with a self-interested buyer is simply not the right market: there should instead be a social planner / central third party mediator (government / regulator) who facilitates the procurement for the buyer and payments to sellers -- perhaps providing a subsidy to the buyer for creating societal value, and maybe taxing sellers for overcharging -- rather than asking the buyer to optimize welfare within the same procurement model.

---

> ### Author Rebuttal · Authors · 2026-03-30
>
> > Q1: The paper motivates utility maximization since unused budget is valuable to the buyer, and yet the formalization seems to discard the very budget surplus that motivates the objective. It seems like Proposition 4 is sensitive to the exact utility formalization, and the paper should explain more carefully why v−p is the economically correct notion rather than v+(B−p), or clarify that the impossibility is tied to this specific formulation, or construct a new example for the v+B-p formulation.
>
> The suggestion to study $v + (B - p)$ instead is very interesting, and our Proposition 4 is indeed sensitive to $v-p$. Specifically, for a single buyer case, which is the case discussed in Prop. 4, there exists a prior-free mechanism that gives a 2-approximation for the proposed objective: whenever $B>v$, do not purchase the item (the optimal solution can obtain at most $v+B < 2B$), otherwise, offer a price $B$ for the seller, which is optimal for this subcase.
>
> We point out that this alternative objective actually has quite different results and economic meaning. If the objective were $v + (B - p)$, then in instances where the budget is larger than the value ($B > v$), a mechanism that purchases nothing could be near-optimal, as reasoned above. In contrast, $v - p$ captures the profit from procurement at a given liquidity level $B$, or alternatively, how effectively the buyer converts budget into value, so these objectives really correspond to different economic questions, despite sharing the same optimal solution.
> This distinction is analogous to the difference between Gains from Trade (GFT) and Social Welfare (SW) in two-sided markets. Both are natural and important objectives, but they emphasize different aspects: GFT measures the improvement generated by trade, while SW includes all endowments and may be maximized even by not allocating. Similarly, our $v - p$ objective isolates the buyer's value created by procurement decisions, whereas $v + (B - p)$ incorporates the buyer's endowment and leads to a different optimization problem.
>
> In summary, adding the budget to the objective is absolutely an interesting but complementary direction. Proposition 4 and Section 4 are tailored to the utility objective $v - p$, and do not directly extend to the alternative formulation. However, as noted, our objective is harder to approximate (much like GFT), so any positive results for it imply positive results for the proposed objective as well. We are happy to revise the paper to clarify this distinction and better position our objective alongside other natural formulations.
>
> > Q2: Although the welfare objective is mathematically natural and gives interesting theoretical results, its economic interpretation in the procurement setting is unclear, and the paper would benefit from a much clearer explanation of who the optimizing principal is supposed to be when welfare is the objective, and why that objective should arise endogenously in this market.
>
> Welfare is commonly chosen as an objective for a mechanism design even when it does not seemingly align with the designer's personal objective. This is because social welfare is the sum of the utility of all participants in the system. For instance, in Google ad auctions, individually Google would like to maximize their revenue. However, because their long-term revenue depends on having participants in the system, they maximize the system's welfare, which accounts for the sum of all participants' utilities (their revenue and the buyers' utility), as this ensures that buyers will stay in the system and continue to participate [Edelman Ostrovsky Schwarz 2007]. The argument holds for procurement auctions as well: while the procurer individually cares about their utility, they may instead choose to maximize the system's welfare for the longevity of the system, to ensure that sellers stay in the system and they can continue to earn utility from them. This is further established by recent work on welfare maximization in procurement (without budgets) authored by a Google research team (Deng et. al. ICML'25 (spotlight)). We can add this discussion to the final version of the paper.

---

> > ### Author Rebuttal · Reviewer_cVmV · 2026-04-04
> >
> > Thanks for the response. I'm happy to increase my score.

---

### Decision · Program_Chairs · 2026-04-30

**Decision:**

Accept (regular)

**Comment:**

This paper studies procurement auctions with budgets and establishes a number of theoretical results both positive and negative results about (approximate) optimality across both prior-free and Bayesian settings.  The final consensus after interaction with the authors is the results do make a sound, sufficient contribution.  The primary remaining weaknesses that limited enthusiasm are that (a) there were lingering concerns about results tightly tied to particular assumptions (e.g. the choice of v-p or budget formulation in Bayesian setting) that limit the range of applicability and (b) there was a desire to see somewhat more technical novelty.

On a presentation note, the references could use some cleanup.  For example Myerson 1981 appears twice, as does "Budget Feasible Mechanisms" (with one reference to an early version with a different author list).